**METHOD**

# MalariaSED: a deep learning framework to decipher the regulatory contributions of noncoding variants in malaria parasites

Chengqi Wang[1*], Yibo Dong[1,2], Chang Li[1], Jenna Oberstaller[1], Min Zhang[1], Justin Gibbons[1], Camilla Valente Pires[1], Mianli Xiao[1,3], Lei Zhu[4], Rays H. Y. Jiang[1], Kami Kim[5], Jun Miao[5], Thomas D. Otto[6], Liwang Cui[5], John H. Adams[1] and Xiaoming Liu[1]

*Correspondence:
chengqi@usf.edu

[1] Center for Global Health and Infectious Diseases Research and USF Genomics Program, College of Public Health, University of South Florida, Tampa, FL, USA
[2] Present address: Florida Department of Health, Jacksonville, FL, USA
[3] Department of Epidemiology and Biostatistics, College of Public Health, University of South Florida, Tampa, FL, USA
[4] School of Biological Sciences, Nanyang Technological University, Singapore, Singapore
[5] Department of Internal Medicine, Morsani College of Medicine, Tampa, FL, USA
[6] School of Infection & Immunity, MVLS, University of Glasgow, Glasgow, UK

## Abstract

Malaria remains one of the deadliest infectious diseases. Transcriptional regulation effects of noncoding variants in this unusual genome of malaria parasites remain elusive. We developed a sequence-based, ab initio deep learning framework, MalariaSED, for predicting chromatin profiles in malaria parasites. The MalariaSED performance was validated by published ChIP-qPCR and TF motifs results. Applying MalariaSED to ~ 1.3 million variants shows that geographically differentiated noncoding variants are associated with parasite invasion and drug resistance. Further analysis reveals chromatin accessibility changes at *Plasmodium falciparum* rings are partly associated with artemisinin resistance. MalariaSED illuminates the potential functional roles of noncoding variants in malaria parasites.

## Background

Malaria remains one of the largest global public health challenges, with an estimated ~ 241 million cases worldwide in 2020 despite remarkable achievements in reducing this deadly disease over the past decade. It has been reported that some noncoding regions in *P. falciparum* are associated with artemisinin resistance (ART-R) [1, 2] or regulate a group of genes related to crucial biological processes, such as antigen variation [3, 4], merozoite invasion [5–7], and gametocytogenesis [6]. Therefore, understanding the effects of genomic variants in malaria parasites is critical for monitoring the spread of drug resistance and the escape of the host immune response. Unlike genomic variants in protein-coding regions whose impacts can often be deduced from resultant codon changes, functional effects associated with noncoding variants cannot be so easily assessed. Despite the large number of noncoding genomic variants recorded in the public variant-tracking resource MalariaGEN [8] (~ 1.31 million noncoding vs. ~ 1.35 million

coding), detecting the contribution of mutations located at noncoding regions is still challenging, and establishing the role of the vast noncoding space in the genetic basis of crucial parasite phenotypes remains difficult.

Genome-wide association studies (GWASs) have been widely used to identify variants in the *P. falciparum* genome significantly associated with drug resistance [1, 9–18]. However, GWAS is an a posteriori approach requiring many historical samples that take time to acquire. Although functional screens have been implemented previously [15, 19], these had limited throughput for testing the whole repertoire of noncoding variants, and these procedures are also time-consuming and expensive [20].

Sequence-based prediction models provide a new perspective on the effects of genomic variants on gene regulation. However, the majority of available models were developed in model organisms [21–24], and their applicability to malaria parasites whose genomes have several unusual features [25] requires careful evaluation. The most striking trait of the *P. falciparum* genome is its high AT content (~ 80% on average, rising to ~ 92% in intergenic regions). Additional features complicating the applicability of existing predictive models include the outsized contributions of a relatively small 27-member sequence-specific transcription factor (TF) family known as ApiAP2s to transcriptional regulation [26]. In comparison, the similarly sized genome of the yeast *Saccharomyces cerevisiae* contains ~ 170 sequence-specific TFs [27]. A sequence-based model accounting for these peculiarities is required for the accurate prediction of gene expression profiles associated with noncoding variants specifically for malaria parasites.

In this work, we have leveraged recent advancements in deep learning (DL) to develop a sequence-based computational framework, MalariaSED, for chromatin profile peak prediction in malaria parasites. To our knowledge, this is the first time in malaria parasites using DL to learn regulatory sequence patterns, including TF binding, chromatin accessibility, and histone modification profiles. MalariaSED predicts the probability of presenting the signal peak of different chromatin profiles and can be applied to predict epigenetic effects based on sequence alterations at single-nucleotide resolution, enabling us to trace the associated biological process.

## Results

### MalariaSED accurately predicts chromatin profiles from genomic sequences in malaria parasites

The unusual genome property (e.g., ~ 80% AT content on average, rising to ~ 92% in intergenic regions) of malaria parasites requires careful evaluation of the existing deep learning approaches for understanding the regulatory sequence code from parasite noncoding sequences. To this end, we developed the first sequence-based deep learning framework, MalariaSED (Additional file 1: Table S1), for malaria parasites, which learns the sequence code of transcriptional regulatory elements by training in large-scale epigenetic experimental data. The data includes four open chromatin accessibility (ATAC-Seq) [28] profiles across *P. falciparum* intraerythrocytic development cycle (IDC), six transcription factors (TFs) with their nine binding profiles during *P. falciparum* and *P. Berghei* IDC and sexual stages, and two H3K9ac histone mark profiles in *P. falciparum*

mid- and late IDC stage [5–7, 29, 30]. MalariaSED computes the probability of presenting the chromatin profile peaks, $f(s)$, for a sequence $s$ using three stages:

$$f(s) = \text{net}(\text{LSTM}(\text{Conv}(s)))$$

Here, $s$ is an input sequence represented as a $4 \times l$ matrix, where each column represents a position in the sequence with a "1" in the row corresponding to the nucleotide at that position and has zeros otherwise (Fig. 1A). Our results indicate that the 1-kb sequence context substantially improves the performance of MalariaSED (Additional file 2: Fig. S1). The convolution layer (Conv) is a motif scanner analogous to a position weight matrix. The bidirectional long short-term memory (LSTM) uses forward and reverse complement features as input to capture the relationship between a set of sequence information outputted from the convolution layer. The LSTM is crucial since TF binding sites may have a group of sequence patterns separated from each other by a long distance. We showcased the significance of LSTM by substituting it with a convolutional layer, and this comparison can be found in the following paragraph. We used the flatten layer and sigmoid function ("net" in the equation) to generate chromatin prediction scores. The high predictive performance for 15 chromatin profiles at multiple parasite stages in both *P. falciparum* and *P. berghei* demonstrates that the MalariaSED architecture can capture important patterns in DNA sequences (evaluation on random splitting strategy, see the "Methods" section for detail, average area under the receiver operating characteristic curve (auROC) = 0.96, area under the precision-recall curve (auPRC) = 0.73, Fig. 1B, Additional file 3: Table S2). We also tested the MalariaSED performance on a chromosome-splitting strategy with specific chromosomes separated as a test set, achieving an average auROC = 0.93 and auPRC = 0.59 (Additional file 4: Table S3). We note that this is the preferred evaluation strategy as it avoids biases that result from potentially overlapping examples [31]. Figure 2 illustrates an agreement between the predicted PfAP2-G binding profile and ChIP-seq/input ratio [6] across a genomic region of chromosome 10 throughout commitment and early

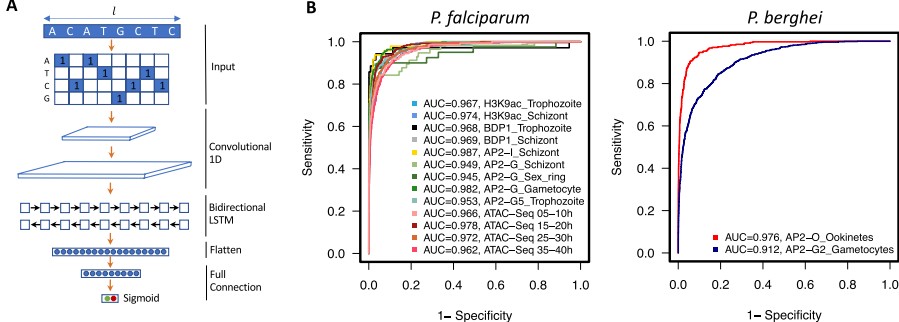

**Fig. 1** MalaiaSED accurately predicts the chromatin profiles in malaria parasites. **A** The general framework of the deep learning model. The input DNA sequence is encoded into the four-row matrix by one hot encoder, where each element represents the appearance of a nucleotide at a specific location. The following two convolutional layers capture the cumulative effects of short sequence patterns by analogy to motif scan. The bidirectional LSTM layer follows up to summarize long-term dependencies between captured DNA patterns. Outputs are fed into the flatten and full connection layers. We calculate the final score by dense layer with sigmoid activation. **B** Performances of the deep learning framework are measured by area under the curve in multiple experimental epigenetic profiles in *P. falciparum* and *P. berghei*

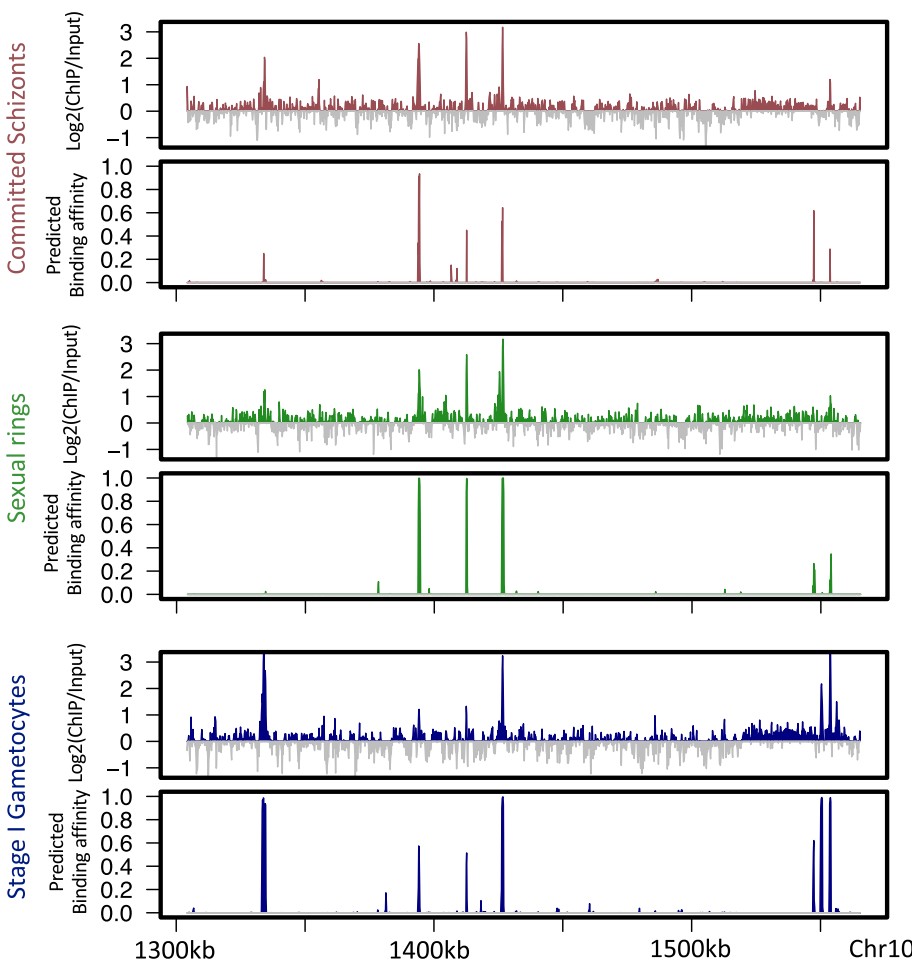

**Fig. 2** MalariaSED can predict the binding dynamic of PfAP2-G throughout commitment and early gametocytogenesis. High agreement between log2-transformed PfAP2-G ChIP/input ratio and predicted genomic tracks from MalariaSED over a region of chromosome 10. The PfAP2-G ChIP intensity covers the parasite stage committed schizonts, sexual ring, and stage I gametocyte

gametocytogenesis. We developed a web interface to facilitate users to easily access MalariaSED (malariased.charleywanglab.org).

### The LSTM layer enhances the performance of MalariaSED in comparison to using solely convolutional layers

We test the DL model with LSTM replacement to a convolutional layer since the convolution DL networks have achieved high prediction accuracy in yeast [22] and the human genome [24, 32]. Furthermore, this would demonstrate the contribution of LSTM in MalariaSED. We set the kernel number of each convolution layer the same as the DL network previously published in the human genome [24, 32] (Additional file 5: Table S4, see the "Methods" section for details). The hyperparameters in the three-layer convolutional network were optimized by Bayes optimization [33] and compared with MalariaSED. The comparisons indicate MalariaSED with LSTM layer achieved better performance in most chromatin profile predictions than the

three-layer convolutional network (average auROC 0.96 vs. 0.93, auPRC 0.73 vs. 0.71 in test set; Additional file 3: Table S2). The only two exceptional cases were PbAP2-O and PfAP2-G in the sexual stages.

### CRISPR/Cas9 mutations in ApiAP2 binding sites demonstrate the capacity of MalariaSED to identify damaging genetic mutations

Unidirectional flow from sequence information to consequent epigenetic status in the MalariaSED framework provides a novel technique for estimating the noncoding variant effects in *malaria* genomes. We tested this capability in predicting ChIP profile changes at multiple TF binding motif mutations generated by the CRISPR–Cas9 system. These ChIP-qPCR profiles are currently available for three TFs, PfAP2-G5 [34], PfAP2-G [6], and PfAP2-I [5], in *P. falciparum*. Four PfAP2-G5 motif mutations, S1-mutation (mut), S2-mut, S3-mut, and S3-deletion (del), were generated at the *ap2-g* gene upstream regulatory region [34]. Their surrounding [− 100, + 100 bp] regions are illustrated in Fig. 3A and as S1, S2, and S3. The PfAP2-G5 ChIP-Seq results indicate a strong binding affinity of PfAP2-G5 at S1 and S2, whereas S3 presents a moderate level. The mutations S1-mut and S2-mut from the original ChIP-qPCR profiles showed minor binding effects, but S3-mut and S3-del strongly weakened PfAP2-G5 binding sites. Further comparison between S3-del and S3-mut indicated higher binding effects at S3-mut. We validated the predictions from MalariaSED by these ChIP profiles. The estimated PfAP2-G5 binding probability from MalariaSED, represented as S1-WT, S2-WT, and S3-WT in Fig. 3B, is

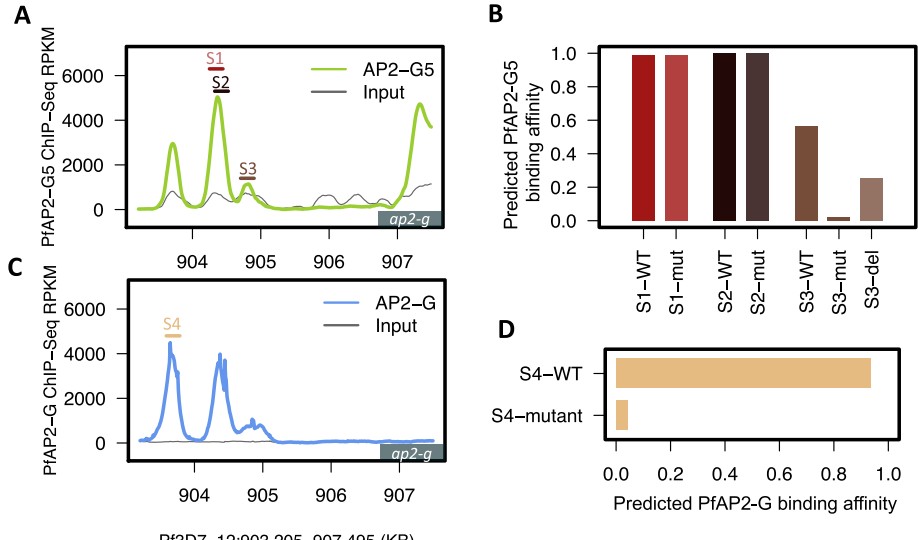

**Fig. 3** PfAP2-G5 and PfAP2-G binding affinity predicted from MalariaSED are consistent with previously published ChIP-qPCR results. **A** PfAP2-G5 ChIP-Seq intensity along the promoter region of *ap2-g* during the trophozoits stage. ChIP-seq results come from a recent study [34]. The flanking region [− 100, + 100 bp] of each mutation on PfAP2-G5 binding sites is highlighted as S1 (dark red), S2 (black), and S3 (brown). **B** The binding affinity predicted from MalariaSED is consistent with the recently published ChIP-qPCR results [34]. The height of each bar indicates the predicted binding affinity of PfAP2-G5 in three mutants highlighted in **A**. **C** PfAP2-G ChIP-Seq intensity [6] along the promoter region of *ap2-g* in early sexual rings. The S4 (dark yellow) indicates flanking region [− 100, + 100 bp] of the mutation of the PfAP2-G motif. **D** Similar to **B**, but showing the predicted binding affinity of the mutant S4 in **C**. The results are in concordance with the published ChIP-qPCR result [6] that PfAP2-G is not able to bind in the mutant

correlated with the ChIP-Seq profiles. In contrast, the binding effects shown as S1-mut, S2-mut, S3-mut, and S3-del in Fig. 3B are in concordance with the original ChIP-qPCR results. We also used MalariaSED to predict the PfAP2-G binding change at a region containing eight GTAC motifs (S4 in Fig. 3C). A previous study showed that mutation of the three motifs by CRISPR/Cas9 significantly reduced the binding of PfAP2-G [6]. This result supports the predictions from MalariaSED, which shows a strong decrease in PfAP2-G binding probability in Fig. 3D. An exceptional case for MalariaSED application in TF binding prediction occurred at the promoter region of the *pfmsp5* gene, where ChIP-qPCR in the original publication showed PfAP2-I binding but exhibited weak binding signal in the ChIP-Seq result [5]. Since MalariaSED was trained in the ChIP-Seq signal, it only captured the sequence patterns that contribute significantly to the high profile of the ChIP-Seq signal. The MalariaSED was not able to predict a PfAP2-I binding in this case with a probability lower than 0.1 for both the WT and ATGCA motif mutations.

### Single-nucleotide substitutions at the TF motif weaken TF binding

The successful validations of TF binding changes in CRISPR/Cas9-generated motif mutations imply potential DNA motif contributions to MalriaSED prediction. To validate this globally, we used MalariaSED to interrogate the average TF binding effects of all possible single nucleotide substitutions (SNSs) at the reported motifs (Additional file 6: Table S5, see the "Methods" section for definition). Basically, for each site of a reported motif, the reference nucleotide was in turn changed by all other three nucleotides as motif mutation sets (e.g., a DNA motif comprising 5 nucleotides generates $5 \times 4 = 20$ motif mutations). Each generated sequence in the mutation sets was individually selected to replace the TF motifs that appeared in all the predicted TF binding regions. The average binding effect of all these SNSs was used to evaluate the motif contribution to TF binding. All other *k*-mer sequences with the same length as the investigated motif were used as the control sets (shown as "ALL" in Fig. 4). Additional file 7: Table S6 displays the average binding effects of all sequences with the same length for each investigated motif. We used this strategy to check the reported motifs of the parasite-specific TF PfAP2-G, a key determinant of sexual commitment in *P. falciparum*. We observed a significantly reduced binding affinity of PfAP2-G after incorporating SNSs at its binding motif GTRC (committed schizonts) and GTAC (sexual rings and stage I gametocytes) (Fig. 4A, Wilcoxon test $p < 2.2e-16$). Similarly, for PfAP2-I, a key regulator of red blood cell invasion, SNSs at the reported motif (GTGCA) significantly weakened its binding affinity at the schizont stage (Fig. 4B, Wilcoxon test $p < 2.2e-16$). Previous results indicate that two motifs (GGTCG and CTTGC) at other DNA-binding domains of PfAP2-I are not required for PfAP2-I binding during the RBC stage [5]. MalariaSED prediction also showed that SNSs in these two motifs did not significantly reduce PfAP2-I binding (Fig. 4B). When another invasion regulator, PfBDP1, was examined, SNSs introduced into the PfAP2-I GTGCA motif led to a significantly reduced binding level in schizonts (Fig. 4C, Wilcoxon test $p < 2.2e-16$), supporting the conclusion that PfBDP1 is likely to form a protein complex with PfAP2-I to regulate invasion genes [5, 7, 35]. For the parasite trophozoite stage, we also checked the DNA binding change of PfBDP1 by introducing SNSs at the PfAP2-I motif GTGCA (Fig. 4C) and did not observe significant changes compared with other 5-mer sequences.

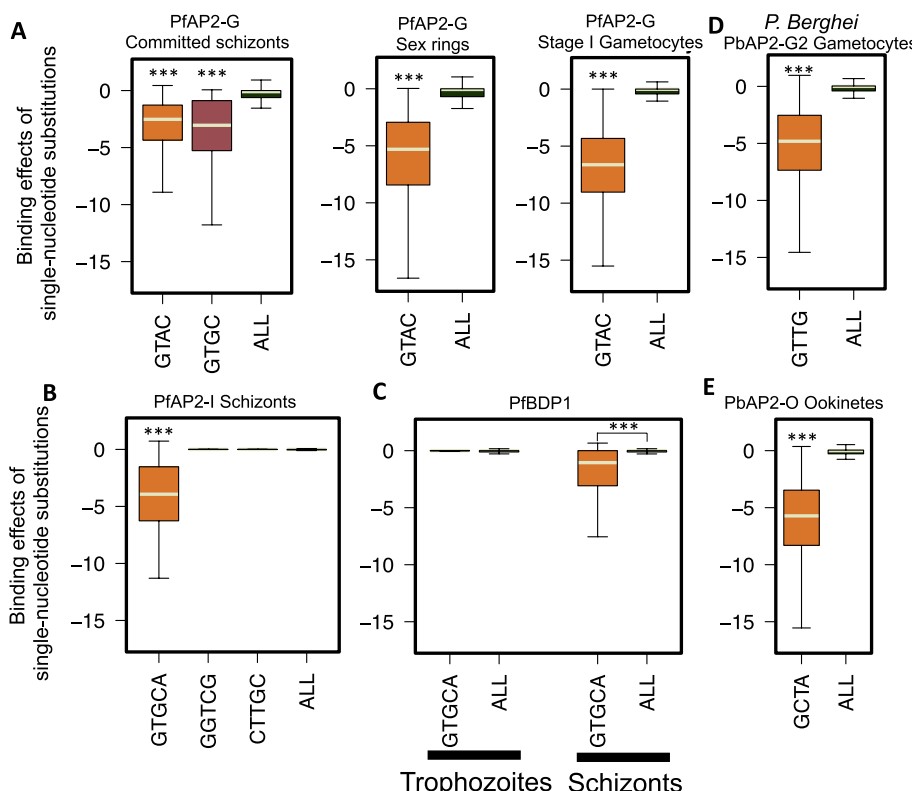

**Fig. 4** TF binding effects of genomic variants predicted by the DL framework demonstrate the contribution of previously identified sequence motifs. **A** Potential single-nucleotide substitutions (SNSs) at reported pfAP2-G motif, GTRC (committed schizonts), and GTAC (sexual rings and stage I gametocytes) significantly reduce the binding affinity of PfAP2-G. Comparisons are between reported PfAP2-G motifs and all 4-mers (ALL) appeared on predicted ApAP2-G binding sites. **B** The PfAP2-I binding effects of base substitutions at the reported motif GTGCA are significant compared with all 5-mers (ALL) in *P. falciparum* genome during schizonts. The other two motifs (GGTCG and CTTGC) from PfAP2-I binding domains 1 and 2 are not able to show significantly reduced PfAP2-I affinity, which is consistent with experimental results [5]. **C** The significantly reduced binding affinity of SNSs at reported PfBDP1 binding motif GTGCA in schizonts supports the previous conclusion that PfAP2-I and PfBDP1 may form a protein complex to bind DNA. However, SNSs at the GTGCA motif are not likely to disrupt PfBDP1 binding at the trophozoite stage. **D**, **E** Single-nucleotide substitutions at two reported binding motifs in *P. Berghei* significantly weaken PbAP2-G2 and PbAP2-O binding sites. ***Wilcoxon test compared with whole-genome background $p < 2.2e-16$

This result supports that PfBDP1 binds many gene promoters not bound by PfAP2-I in trophozoites and is in line with the conclusion that PfBDP1 and PfAP2-I are involved in a protein complex in trophozoites but not in schizonts [7]. Furthermore, the predicted results from MalaraSED do not show significant binding effects of the motifs TAACT and ACAAC on PfBDP1 binding sites (Additional file 2: Fig. S2). It has been shown that TAACT, the second motif of PfBDP1 binding, co-occurs with the micronemal DNA motif ACAAC in the promoters of several micronemal and rhoptry neck genes [5]. The limited contributions of these two motifs suggest that some other unidentified TFs may recruit PfBDP1 to regulate micronemal gene expression. We also tested MalariaSED on two AP2 family TFs, PbAP2-G2 and PbAP2-O, in *P. berghei* [29, 30]. PbAP2-G2 represses asexual genes to support conversion to the sexual stage, while AP2-O plays a critical role in regulating mosquito midgut invasion. MalariaSED identifies SNSs in two binding motifs, GTTG for PbAP2-G2 and GCTA for PbAP2-O, which are more likely to abolish their binding (Fig. 4D, Wilcoxon test $p < 2.2e-16$). Taken together, these results suggest that

MalariaSED can capture crucial DNA sequences responsible for TF binding. Critically, MalariaSED predicted TF binding disruptions at TF binding motifs in mutants from sequence data alone, demonstrating the utility of sequence-based models for interpreting functional variants in malaria parasites.

### Geographically differentiated variants at noncoding regions are more likely to alter their surrounding chromatin environments

Approximately six million curated genomic variants (including SNPs and short indels) from 7000 *P. falciparum* samples recently made available on MalariaGEN show the enrichment of geographically differentiated SNPs in nonsynonymous variants [8]. Further analysis indicates that genes harboring the top geographically variable nonsynonymous SNPs tend to be involved in the process of parasite transmission by the mosquito vector and are associated with drug resistance. Despite noncoding variants showing the second highest geographic differentiation after nonsynonymous variants, the limited functional annotation of malaria parasite noncoding regions restricts our understanding of noncoding-variant contributions to this process. Our initial analysis of ~ 1.3 million noncoding variants [8] (trimming off ~ 1.4 million noncoding loci with low Low_VQSLOD provided by MalariaGEN) indicates that noncoding variants with high levels of geographic differentiation (global $F_{ST} > 0.1$) are located closer to their nearest gene (Fig. 5A, Wilcoxon test $p < 2.2e - 16$ compared with genomic variants showing

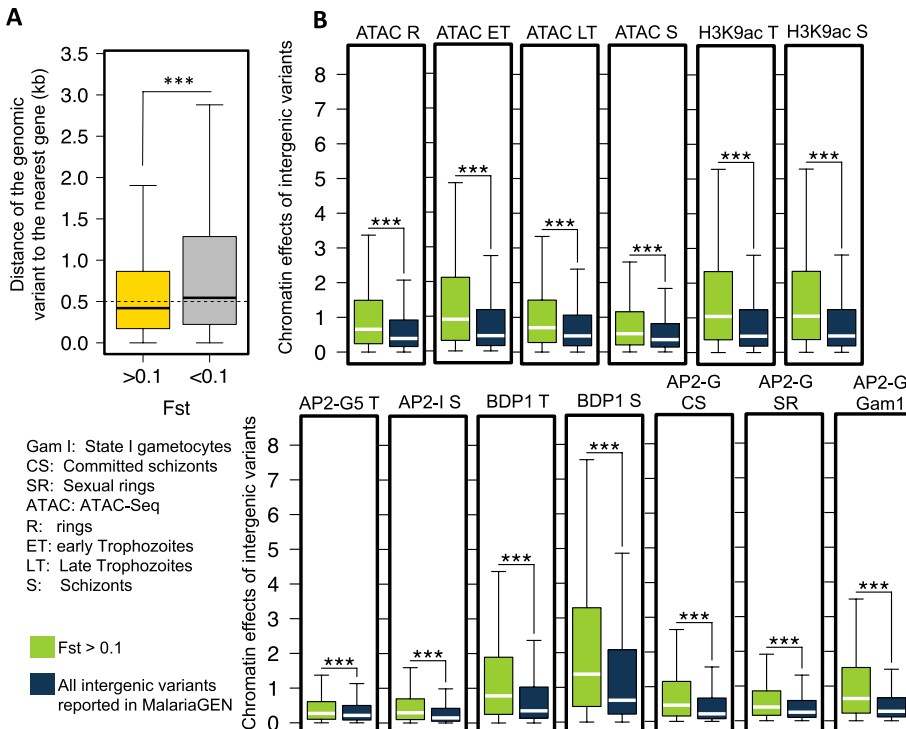

**Fig. 5** Noncoding variants with high levels of geographic differentiation tend to alter their surrounding chromatin profiles. **A** Noncoding variants strongly differentiated at the global level are closer to their neighboring genes. We used global Fst > 0.1 to identify variants with high levels of geographic differentiation. ***Wilcoxon test compared with the remaining noncoding variants, $p < 2.2e - 16$. **B** Noncoding variants that acquired high geographic differentiation (Fst > 0.1) are predicted to bring in significant epigenetic changes at their ± 100-bp surrounding regions

global $F_{ST} < 0.1$), suggesting that the differentiation process at the noncoding region may partly be due to the selection of regulatory activity. We used MalariaSED to predict the impact of chromatin profiles of noncoding variants (Additional file 2: Fig. S3, Additional file 8: Table S7). We observed significant alteration of chromatin profiles in the group of genomic variants presenting strong geographic differentiation (Fig. 5B, Wilcoxon test $p < 2.2e-16$ compared with noncoding variants showing limited geographic differentiation). The higher chromatin effects observed at these geographically differentiated variants indicate that *P. falciparum* may use epigenetic mechanisms to locally regulate the expression of genes of certain biological pathways to benefit its survival.

### Drug resistance and red blood cell invasion-related pathways are associated with geographic differentiation in P. falciparum noncoding regions

We next investigated the biological pathways associated with strong geographic differentiation at noncoding regions by assigning variants to their nearest genes ($F_{ST} > 0.1$, distance to the nearest gene < 3 kb). Only the noncoding variants leading to significantly predicted chromatin profile changes (the top 1% of chromatin effects) were selected. Pathway enrichment analysis indicated that oxidative stress defense (S-nitrosylation and S-glutathionylation), proteasome degradation (ubiquitin-protein conjugates), and food vacuoles were associated with noncoding variants predicted to cause significant chromatin accessibility changes in the ring and early trophozoite stages (Fig. 6A, B, Additional file 9: Table S8 for the complete list of enriched pathways). Other enriched pathways included protein transport (PTEX export), epigenetic

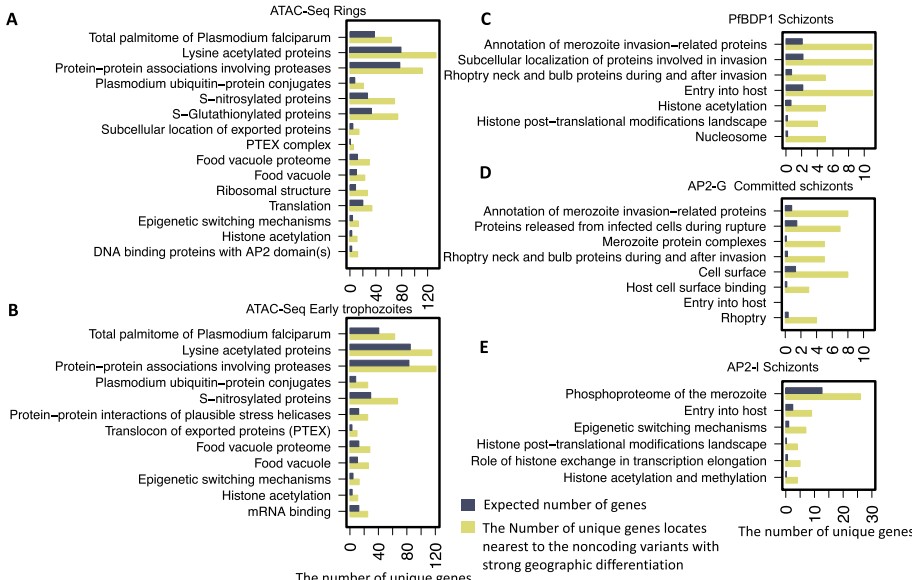

**Fig. 6** The enriched functions or pathways upon noncoding variants with strong geographic differentiation. The height of the yellow bar indicates the number of genes assigned to noncoding variants that are strongly geographically differentiated and significantly change the chromatin profile. To estimate the number of genes expected to observe for each function or pathway term, the gene fraction in the *P. falciparum* genome presenting the specific functional term is multiplied by the number of assigned genes to geographic differential variants exhibiting the high chromatin effects (hypergeometric test, FDR-corrected *p*-value < 0.1, check Additional file 9: Table S8 for full list)

modifications, and the palmitome. Notably, most of these pathways have been previously demonstrated to be associated with altered transcription profiles in ART-R parasites [36, 37]. Our results suggest that geographically diverse noncoding variants may enable the parasite to modulate the transcription of key pathways underlying the artemisinin response, ultimately allowing survival under drug pressure. Interrogating enriched genes around each geographically differentiated variant leading to binding affinity changes for the TFs PfBDP1, PfAP2-G, and PfAP2-I (Fig. 6C–E), we identified a group of genes related to merozoite invasion, including *EBA181, AARP, RhopH2, RhopH3* and *SRA, RON2, MSP9, RAP1.* PfBDP1 is a chromatin-associated protein responsible for regulating the expression of genes involved in red blood cell (RBC) invasion [7]. TF PfAP2-I has been shown to associate with PfBDP1 and positively regulate the transcription of invasion-related genes [5]. As a known pivotal activator of early gametocyte genes, PfAP2-G was also recently reported to have a potential role in regulating genes important for RBC invasion [6]. *P. falciparum* strains present broad diversity in invasion phenotypes and gene expression across populations [38]. The TF binding affinity changes surrounding noncoding variants upstream of invasion-related genes may explain how parasites maintain this diversity of gene expression.

### Significant chromatin accessibility changes surround artemisinin resistance-related eQTLs

The large-scale expression quantitative trait loci (eQTL) and transcriptome-wide association studies (TWAS) study conducted by Tracking Resistance to Artemisinin Collaboration I (TRACI) identified more than 13,000 SNS-expression linkages in 773 parasite isolates from the Greater Mekong Subregion (GMS) [1]. The result provides a new perspective to understand ART-R associated with SNSs through the unidirectional flow of information from sequence changes to consequent gene expression changes and, ultimately, phenotype alteration. However, the extensive gaps in our understanding of genomic changes and the resulting transcriptional alterations in malaria parasites impede the use of TWAS. Considering the correlations between chromatin accessibility and neighboring gene activity [28], we reasoned that the surrounding chromatin profile changes resulting from eQTLs would lead to gene transcription changes. To test the associations between the eQTL and its surrounding chromatin profiles, we investigated the capacity of different chromatin profile effects to distinguish local intergenic eQTLs from SNS reported in MalariaGEN [8]. Our results show that chromatin accessibility in rings outperforms other epigenetic markers in different parasite stages as an indicator/predictor of eQTLs (auROC = 0.7, Fig. 7A, Additional file 10: Table S9), consistent with skewness toward the ring-stage parasites collected in the TRACI study [1]. We further tested this on a group of local eQTLs linked with ART-R by regulating ART-R-related gene expression (see the "Methods" section). Ring-stage chromatin accessibility also presented the highest predictive performance in identifying ART-R-associated eQTLs (auROC = 0.73, Fig. 7B), indicating that transcriptional regulatory activity changes may lead to neighboring gene transcriptional alterations associated with ART-R phenotypes. Taken together,

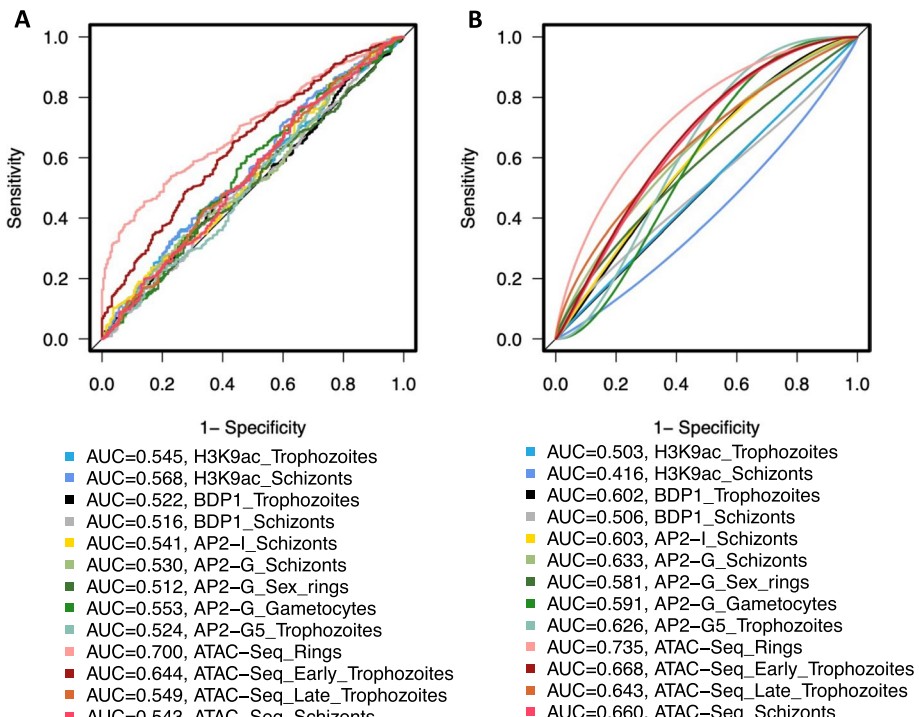

**Fig. 7** Chromatin accessibility alternation is surrounding previously identified eQTLs in the TRACI study. **A** The chromatin effects of reported eQTLs in the TRACI study are used to discriminate them from all SNPs gathered in MalariaGEN. **B** The eQTLs linked with ART-R-associated transcriptions were also investigated. The eQTL prediction performance of different chromatin profiles is measured by area under the curve

these results demonstrate that MalariaSED is informative for predicting the transcriptional impact of common noncoding variants in *P. falciparum*, such as eQTLs.

## Multiple chromatin profile alterations at the schizont stage show signatures of intergenic selection

Previous analysis of single-cell sequencing data revealed strong evidence that de novo mutations are under selection in *P. falciparum* populations [39]. This evidence includes the skewness toward protein-coding regions, high nonsynonymous vs. synonymous substitution ratio, and elevated expression levels of de novo mutations-targeted genes. Because of the limited functional annotation of *P. falciparum* intergenic regions, this analysis was focused on coding regions. We used MalariaSED to investigate the chromatin effects of the 35 noncoding de novo mutations reported in the SCS study. Compared with the genome-wide background distribution generated from all SNSs reported in MalariaGEN [8], significant changes were observed in schizonts, including chromatin accessibility, H3K9ac, and two TFs, PfAP2-I and PfAP2-G binding affinities (Fig. 8, Wilcoxon test $p < 0.05$, Additional file 2: Fig. S4 for other chromatin profiles). Our results show signatures of selection for intergenic variants in schizonts, which could result in neighboring gene expression changes.

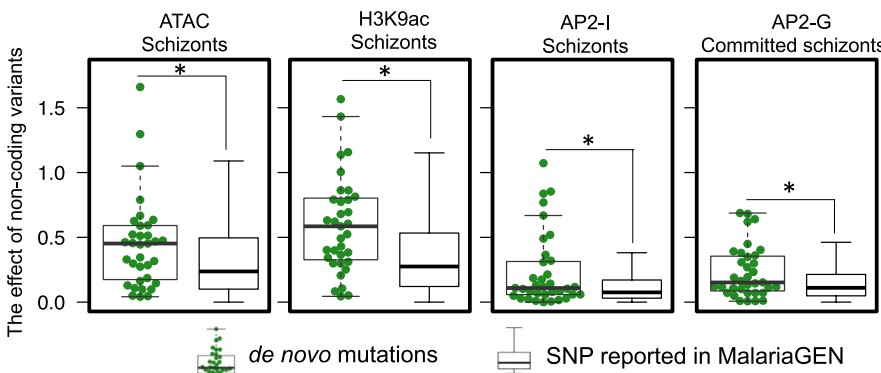

**Fig. 8** The results from MalariaSED indicate de novo mutations at noncoding regions discovered from the single-cell study have a higher chance to alter their surrounding chromatin profiles in schizonts. *Wilcoxon test compared with noncoding SNPs collected from MalariaGEN, $p < 0.05$

## Discussion

This work presents the first sequence-based computational framework, MalariaSED, that learns regulatory sequence codes in malaria parasites. The DL-based framework enables us to predict the chromatin effects of sequence alterations with single-nucleotide sensitivity, thereby providing a means to assess the functional relevance of noncoding variants. We demonstrated this by using previously published ChIP–qPCR data from CRISPR/Cas9-generated mutations in AP2 TF-family binding motifs [6, 34]. Nonetheless, one exceptional case occurred at the *pfmsp5* promoter, which was not identified in PfAP2-I ChIP-seq but was detected in the ChIP–qPCR experiment [5]. MalariaSED could not predict PfAP2-I binding in either the wild type or parasites carrying mutations at the PfAP2-I binding motif at the *pfmsp5* promoter. We reason that some DNA sequence patterns may be related to discrepancies between the ChIP–qPCR and ChIP-Seq results. This DNA sequence pattern is reflected at the *pfmsp5* promoter and cannot be captured by MalariaSED trained in ChIP-seq data. We also demonstrated the significant contributions of previously identified DNA motifs to known TF binding, such as the members in the PfAP2 family, PfBDP1, PbAP2-O, and PbAP2-G2 [5–7, 29, 30]. We are not clear on the reasons for MalariaSED not capturing previously reported PfAP2-G5 binding motifs, although it achieves prediction performance auROC > 0.95 and validates previous CRISPR/Cas9 results. One explanation is that MalariaSED may use DNA patterns far apart from each other in *P. falciparum* genome for AP2-G5 binding predictions.

We used MalariaSED to explore the epigenetic effects of large amounts of noncoding variants released from MalariaGEN partner studies [8]. The results indicate that noncoding variants that acquire high levels of geographic differentiation significantly likely impact the surrounding epigenetic profiles. This suggests that some noncoding regions have been at least in part subjected to natural selection, which is consistent with the previous analysis results on coding regions. It is unclear why all epigenetic markers present significant changes surrounding geographic differentiation variants. Further study is needed to understand the dependence of each chromatin profile. An alternative explanation is deleterious changes that have spread to high frequency due to hitchhiking with selected coding variants. We test one of the consequences of this argument, which is that geographically differentiated variants accompanied by different chromatin

profile effects in the noncoding region are more likely closer to genes with high levels of geographic differentiation. The analysis was based on the global Fst score calculated by MalariaGEN [8] for each gene based on the highest-ranking non-synonymous variants within the gene. We observed that geographically differentiated noncoding variants with higher alternation in AP2-G binding profile in commit schizonts, AP2-G5 in trophozoites, chromatin accessibility, and euchromatin tend closer to the gene harboring highly differentiated non-synonymous variants (Additional file 2: Fig. S5, Wilcoxon test $p < 0.05$). The results imply that the non-synonymous variants with high geographic differentiation in the malaria parasite could hitchhike neighboring noncoding variants to enhance or repress the gene expression by altering the chromatin compactness. We speculate that some histone modifications may recruit specific TFs and that the binding affinity alterations may result from these marker changes. Our reanalysis of ~ 1.3 million noncoding variants also provides new insight into parasite local geographical adaptation driven by noncoding variations. The associated potential transcriptional change in protein trafficking, epigenetic and translational machinery, and the alternations of many PfAP2-G binding affinities provide new evidence that parasites trade off reproduction (transmission to new hosts) and growth (within-host replication) to adapt to different local regions where parasites present diverse transmission intensity [40]. We need to point out that the PfAP2-G binding change regions tend to be located upstream of genes important for RBC invasion. This is further demonstrated by the overlapping gene list with PfAP2-I and PfBDP1, as well as the limited functional enrichment in sexual rings and stage I gametocytes for PfAP2-G binding site changes. We also observed that a group of geographic differentiation variants located at noncoding regions are likely to regulate gene functions in food vacuoles, oxidative stress defense, and proteasome degradation. This reflects the evolution of genetically distinct *P. falciparum* subpopulations that are thought to occur due to high drug pressure and oxidative stress [1, 41]. The subsequent step involves assessing the fitness cost of these noncoding variants. It is plausible that these variants may exert mild deleterious effects, contributing to resistance or other potential advantageous phenotypes for parasite growth. To explore this further, a large-scale genetic screen is required to investigate the reported noncoding variants under the parasite's normal growth conditions. Such investigations would provide valuable insights into the functional implications of these variants and their impact on the overall fitness of the parasite.

To test the epigenetic contributions resulting from intergenic variants to transcriptional change, we used chromatin effects to discriminate reported eQTLs [1] against the nontrait-associated SNSs collected in MalariaGEN partner studies [8]. We observed high concordance between the chromatin accessibility change at the ring stage and the local eQTL discovered in *P. falciparum* clinical isolates. This result is also shown in the subset of eQTLs linked with ART-R via transcriptional profiles. The agreement between the clinical isolate stage (high skewness toward rings) and the ring stage of the chromatin profile supports a previous study showing that the chromatin accessibility pattern correlates with neighboring gene expression values [28]. Furthermore, MalariaSED bridges the gap in our understanding between DNA sequence changes and the resulting epigenetic profile alterations in malaria parasites. Building on the known association between gene transcription and parasite phenotype,

the directional flow of information from sequence variation to consequent chromatin profile change to gene transcription alternation and further to ART-R effect will allow us to extract the causality of variant(s). Based on currently available chromatin profiles, we are not able to identify any chromatin profiles associated with distal eQTLs (auROC < 0.65 for all 13 chromatin profiles here). This may be due to limited epigenetic data available in *P. falciparum* and the fact that only one chromatin profile in rings can be implemented in MalariaSED. It will be interesting to study these distal eQTL hotspots when new epigenetic data are available for MalariaSED training.

MalariaSED facilitates the functional explanations of de novo mutations discovered in single cell sequencing studies. The significant de novo mutations enrichment in coding regions and the disproportionate genes targeted by de novo mutations suggested strong selection in the *P. falciparum* population collected in Chikhwawa, Malawi [39, 42]. This includes the ApiAP2 gene family, implying selection in the epigenetic regulation mechanism. We used de novo mutations to investigate the chromatin effects of intergenic de novo mutations and identified significant changes in schizonts, including H3K9ac, PfAP2-G, and PfAP2-I binding affinities, as well as chromatin accessibility. However, due to the limited number of intergenic de novo mutations in the single-cell sequencing study, we are hesitant to make conclusions about potential functions associated with these 35 noncoding de novo mutations. Further study is required when more single-cell sequencing datasets are generated.

MalariaSED proved to be a successful application in malaria parasites, shedding light on the potential regulatory changes arising from noncoding variants. However, to enhance and validate MalariaSED's performance, additional endeavors are necessary to integrate it with established DL frameworks in model organisms [21, 24, 32]. We conducted a comparison with a convolutional network that employed the same number of kernels as the model previously developed for the human genome [24, 32]. While we optimized the hyperparameters for this convolutional network, further investigation is required to examine the impact of different kernel sizes on the prediction performance. Moreover, it is crucial to explore the potential application of sophisticated DL models, such as Enformer [21], in the context of malaria parasites. In our study, we conducted tests using various sequence inputs for Enformer and observed promising results, achieving an overall Pearson correlation of 0.63 (Additional file 11: Table S10). To fully unlock the potential of Enformer in malaria parasites, further efforts are required to optimize its hyperparameters. Fine-tuning the model's parameters can potentially lead to improved performance and more accurate predictions.

We need to notify our training process is based on randomly splitting 200-bp intergenic genome segments into training/validation/test sets. Since the extension of each input 200-bp sequence to 1 kb, this may bring in the sequence in the training set had highly overlapping regions with the test set, thereby inflating the test results. To break the dependence between samples in training, validation, and test sets, we used a chromosome splitting strategy to split samples such that all samples on the same chromosome were all in one of the training, validation, and test sets. We used the same training process for both the random splitting and chromosome splitting data set The results show an average 0.03 decreased performance in auROC (0.96 vs. 0.93) and 0.14 decrease in auPRC (0.73 vs. 0.59, the former indicates random splitting) compared

with the random splitting data set (Additional file 4: Table S3). The results indicate the importance of properly evaluating the chromatin profile prediction method by rigorously segregating the training, validation, and test sets.

In the long run, we expected that sequence-based regulation analysis would become a crucial part of malaria parasite research, especially to help unveil the regulatory activity of noncoding genomic regions, which are currently poorly understood in malaria parasites. Such analyses could, in the future, be implemented to prioritize noncoding variants potentially perturbing gene expression contributing to parasite immune escape and drug resistance. With the increasing data availability of functional variants, the approach can be readily trained, adapted, and further improved.

## Conclusions

We have effectively showcased the utilization of a deep learning framework for predicting chromatin profiles in the atypical genome of malaria parasites. Our developed framework, MalriaSED, has been utilized to investigate the epigenetic regulatory implications of reported noncoding genetic variations. Our findings provide evidence suggesting that the malaria parasite may potentially exploit this resulting regulatory adaptability as a survival strategy to escape the immune system and resist drug pressures. MalariaSED offers an efficient and accessible platform that can facilitate further research into unraveling the mechanisms underlying noncoding genetic effects in malaria parasites. We anticipate that MalariaSED will play a pivotal role in uncovering regulatory insights within the extensive, yet poorly understood, noncoding regions of the parasite's genome. Furthermore, it stands to contribute significantly to our understanding of the mechanisms enabling drug resistance and immune evasion in these parasites.

## Methods

### Datasets for MalariaSED development

The previously published ATAC-Seq datasets [28], nine ChIP-seq datasets, including PfAP2-I [5], PfAP2-G [6], PfAP2-G5 [34], PfBDP1, and histone modification H3K9ac [7], from *P. falciparum* RBCs to the gametocyte stage and two ChIP-Seq datasets for PbAP2-O and PbAP2-G2 [29, 30] in *P. berghei*, were used to develop the MalariaSED framework. The whole genome is divided into bins of 200 bp, yielding more than 50,000 bins not overlapped with the coding regions. A bin is labeled "positive" if more than 60% of it overlaps with ChIP or ATAC peaks, while the remaining bins are used as negative datasets. We used the ratio 7/1.5/1.5 to divide all extracted bins into training, test, and validation sets. The number of bins used for MalariaSED training for different epigenetic markers is listed in Additional file 3: Table S2. MalariaSED was also trained and tested in a chromosome-splitting strategy, where bins on the same chromosome were all assigned to one of the training, validation, and test sets. We select chromosome combinations in the training, validation, and test sets so that the ratio is close to 7/1.5/1.5 for the chromatin profiles with low positive samples (positive bin number < 1100). For the remaining chromatin profiles, bins on chromosomes 10 and 11 were selected as the validation set and chromosomes 9 and 12 as the test set. The bins in all other chromosomes were used as a training set (Additional file 4: Table S3).

**Deep neural network models with long short-term memory (LSTM) layers (MalariaSED)**

For each bin 200 bp in length, we extended it by 400 bp in both directions, resulting in a 1-kb input sequence. The 200-bp bin and its surrounding 1-kb sequence context have been successfully used to predict chromatin profiles in the human genome [32, 43]. We used one-hot encoding to convert each 1-kb sequence into a $1000 \times 4$ bit matrix, where "1" represents a specific nucleotide present and "0" does not. The architecture of the MalariaSED framework is shown in Fig. 1A. Two 1D convolutional layers followed up with max pooling and drop-out layers are combined with bidirectional LSTM to capture dependencies of DNA sequence patterns contributing to specific epigenetic markers. Regularization and constraint are applied to all layers to minimize overfitting. After the flatten layer, the final predictions are computed through sigmoid activation in dense layers. The Adam optimizer was used to search for model parameters based on binary cross-entropy. We use early stopping to halt the parameter optimization process if there is no improvement in validation auPRC for ten consecutive epochs. Bayesian optimization [33] was used to optimize the hyperparameters objective to maximum auPRC. Additional file 1: Table S1 lists the range of all hyperparameters in Bayesian optimization. MalariaSED was trained independently in 15 chromatin profiles from *P. falciparum* and *P. Berghei*. MalariaSED was implemented in TensorFlow (v2.4.0) and was trained on 4 GPU Tesla V100 with batch size 2000. The best model was determined by the highest auPRC in the validation set and evaluated by the test set. The evaluation results are listed in Additional file 3: Table S2 and Additional file 4: Table S3. The ROC and PR curves were generated based on the MalariaSED prediction from the network output and the label assigned for each 200-bp bin.

**Deep neural network models with three-layer convolutional network**

We used a three-layer convolutional network to demonstrate the outperformance of MalariaSED in the malaria parasites to the prediction frameworks developed in model organisms [22, 24, 32]. We replaced the LSTM layer in MalariaSED with a convolutional layer using kernel number 960. This can also provide evidence of the contribution of the LSTM layer. We set the kernel numbers of the first and second convolutional layers as 320 and 480. All kernel number was set the same as the DL network previously published in the human genome [24, 32]. Additional file 5: Table S4 provides all hyperparameters considered in the Bayesian optimization. All hyperparameters' search range is the same as the convolutional layer and dense layer in MalariaSED. The same training, validation, and test sets in MalariaSED development were used to evaluate the performance of the three-layer convolutional network. We use early stopping to seize the parameter optimization process if there is no improvement in validation auPRC for ten consecutive epochs.

**Enformer application in the malaria parasite**

We used Enformer to predict 13 chromatin profiles in *P. falciparum*. We select the input length 12.8, 32, and 64 kb to optimize the input sequence length. At least a threefold shorter input sequence was used here since ~22.9 mb length of *P. falciparum* genome, which is 290-fold shorter than the human genome. Also, it has been demonstrated the lack of long-range interactions in *P. falciparum* genome [44, 45]. We set Enformer to

take input with $l \times 4$ tensor, where $l$ represents the input sequence length, and number 4 indicates one-hot-encoded DNA sequence. The input sequence $l(l = 12.8, 32, 64kb)$ was reduced to a $l/128$ positions by the convolutional blocks in Enformer, where each position represents a 128-bp sequence. After the transformer blocks, the cropping layer in Enformer trims half of the positions from both sides, resulting $l/128/2$ positions. Additional file 11: Table S10 lists the number of training, validation, and test sets for different input $l$. We incorporate 13 chromatin profiles in *P. falciparum* genome for Enformer training, indicating the output shape $l/128/2 \times 13$ from Enfomer. The Enformer has trained 1000 epochs for the step same as the training sample number, meaning each epoch goes through all training samples. The selected training model with the highest Pearson correlation in the validation set and the Pearson correlation results in the test set is listed in Additional file 11: Table S10.

### Sequence pattern contribution to MalariaSED

To estimate the sequence contributions to MalariaSED predictions, we predict chromatin profile changes by generating all possible SNSs along with each site of the *k*-mer sequence. Each 200-bp region predicted as positive from MalariaSED was extended by 400 bp in both directions, resulting in a 1-kb sequence window. SNS generation was then performed for every positive 1 kb bp region covering all 4-mer or 5-mer sequences. For each 4/5-mer sequence, we took the average chromatin effects resulting from all SNSs as the contribution to MalariaSED. The effect of an SNS on a specific chromatin marker was measured by the absolute value of log2-fold change of odds predicted from MalariaSED. This measurement was also successfully utilized to evaluate the chromatin effects of genomic variants in the human genome [24, 32].

### Functional enrichment for geographic differential variants at intergenic regions

The intergenic variants presenting high geographic differential (Fst > 0.1) from Malaria-GEN partner studies [8] were selected for this analysis. We further picked up the geographic differential variants predicted to change the particular chromatin profile at the top 1% level to explore the potential biological function change. A gene is used for functional enrichment analysis if its distance to at least one selected variant using the above-mentioned criteria is shorter than 3 kb. To identify GO terms enriched in the gene list of interest, Fisher's exact test was used to test whether more genes with this GO term were observed compared with the background comprising the whole genome.

### Chromatin profile prediction surrounding eQTLs

We downloaded 13,257 eQTLs from the TRACI study [1]. Only local intergenic eQTLs (15 kb between an eQTL and its regulated target(s)) were selected for the positive dataset in DL predictions. We strengthened the ART-R-associated eQTLs by the cutoff of an absolute value of 0.1 for the adjusted Spearman correlation between the eQTL-regulated transcriptional profile and parasite clearance time. All SNSs gathered in MalariaGene [8] are used as the negative dataset. Each reported eQTL or SNS is extended to include the surrounding 1-kb DNA sequence to fit the input requirement of MalariaSED. The chromatin effects of each variant were measured by the log2-fold change difference of odds, which was previously used in human genetic studies [32].

## Supplementary Information

---

**Additional file 1: Table S1.** MalariaSED model configuration.

**Additional file 2: Fig. S1.** Evaluating the performance of MalariaSED using different lengths of DNA sequence input. **Fig. S2.** TFs binding effects of single nucleotide substitutions at 5-mers sequences predicted by MalariaSED in *P. falciparum*. **Fig. S3.** MalariaSED prediction for epigenetic markers of ~1.3 million variants gathered by Pf3K. **Fig. S4.** MalariaSED prediction results for de novo mutations at non-coding regions discovered from the single cell study. **Fig. S5.** The geographically differentiated variants accompanied by different chromatin profile effects in the noncoding region are more likely closer to genes with high levels of geographic differentiation.

**Additional file 3: Table S2.** The prediction performance of MalariaSED, three-layer convolutional network and Enformer in the independent test sets.

**Additional file 4: Table S3.** The test set prediction performance of MalariaSED in the strategies of random splitting and chromosome splitting.

**Additional file 5: Table S4.** Model configuration of three-layer convolutional network.

**Additional file 6: Table S5.** The mean of average SNS effects for all DNA sequence patterns that share the same length as the reported TF binding motifs were calculated.

**Additional file 7: Table S6.** The average SNS effects at each DNA motif through all predicted TF binding sites.

**Additional file 8: Table S7.** The chromatin effects prediction from MalariaSED for all reported genomic variants collected by malariaGEN.

**Additional file 9: Table S8.** The GO and MPMP enrichment for geographically differentiated variants presenting high chromatin effects.

**Additional file 10: Table S9.** The chromatin effects prediction from MalairaSED for local eQTLs reported in TRACI study.

**Additional file 11: Table S10.** Input, output sequence length of Enformer, and performance in the independent test set.

**Additional file 12.** Review history.

---

### Acknowledgements

We thank the USF Genomics Program Omics Hub for the productive discussion. We express our gratitude to Dr. Zbynek Bozdech's lab at Nanyang Technological University for the invaluable guidance in utilizing the TRAC data. We appreciate MalariaGEN for providing support for using their collected large-scale genomic variants data. We thank Dr. Derek Wildman and Dr. Monica Uddin for their long-time support. We appreciate Dr. Ian Cheeseman from the Texas Biomedical Research Institute for providing the noncoding de novo mutations data. We appreciate Dr. Michael Duffy from the University of Melbourne for the guidance in using their published ChIP-seq data. We appreciate Dr. Shijun Shen and Dr. Qingfeng Zhang from Tongji University for the guidance in using their published ChIP-seq data.

### Peer review information

### Review history

The review history is available as Additional file 12.

### Authors' contributions

W.C. conceived and designed the study. D.Y. developed the website. W.C. and L.C. jointly conducted the training on the deep learning model. W.C., L.C., O.J., Z.M., G.J., P.V.C., and X.M. carried out the bioinformatics and statistical analyses. Z.L. made significant contributions to the design and analysis of the TRACI data. W.C., L.X., J.H.A., C.L., O.D.T., M.J., K.K., and J.H.Y.R. provided supervision for the research. W.C. wrote the original draft. L.C., O.J., J.H.A., C.L., and O.D.T. reviewed and edited the manuscript. W.C. is the first author and also holds the role of senior author.

### Funding

This work was supported by the National Institutes of Health grant R01 AI117017 (J.H.A.) and U19 AI089672 (C.L.). This work was supported by the internal award of the College of Public Health at the University of South Florida.

### Availability of data and materials

MalariaSED was trained in ATAC-Seq profiles (GEO: GSE104075 [28]) and nine ChIP-seq datasets, including PfAP2-I (GEO: GSE80293 [5]), PfAP2-G (GEO: GSE120448 [6]), PfAP2-G5 (GEO: GSE184659 [34]), PfBDP1 and histone modification H3K9ac [7], from *P. falciparum* RBCs to the gametocyte stage and two ChIP-Seq datasets for PbAP2-O (GEO: GSE184659 [29]), and PbAP2-G2 in *P. berghei* (GSE66190 [30]). 1.3 million noncoding variations were collected by MalariaGEN v6.0 [8]. A total of 13,257 eQTLs were downloaded from the TRACI study [1]. Source code for Enformer is available at GitHub https://github.com/google-deepmind/deepmind-research/tree/master/enformer [46].

MalariaSED is implemented in Python. Its source code is available under the MIT open-source license. The source code for running and training MalariaSED models is available at https://github.com/CharleyWang/MalariaSED [47] and Zenodo (https://doi.org/10.5281/zenodo.8336741, [48]).

The MalariaSED web portal for 15 epigenetic profile predictions for malaria parasites is at malariased.charleywanglab.org.

## Declarations

### Ethics approval and consent to participate
Not applicable.

### Consent for publication
Not applicable.

### Competing interests
The authors declare that they have no competing interests.

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

## 