## [**Additional file 12. **Review history. · Genome Biology]

Review History

First round of review

Reviewer 1

Were you able to assess all statistics in the manuscript, including the appropriateness of statistical tests used? Yes: Statistics used were appropriate.

Were you able to directly test the methods? No.

Comments to author:

This is a thought provoking paper that uses a deep learning approach to predict the phenotypic impact of non-coding mutations. This is done by training a model, using existing experimental epigenetic data for Plasmodium. The approach is of specific interest to those working on malaria parasites and of broad general interest to those working with a wide range of organisms.

The paper has several strong features:

1. This is extremely important topic, given that there are quite similar numbers of coding and non-coding variants in the Plasmodium genome, and only coding variants have receive significant attention. Furthermore, existing models for making similar predictions in other model organisms work poorly in Plasmodium, most likely because *P. falciparum* is extremely AT biased, and markedly different from most other organisms in having rather few transcription factors.
2. The development and validation of the malariaSED tool is carefully done, with careful optimization of the size of DNA windows for maximal performance, and evaluation of predictive power by showing: (i) Accurate prediction of ApiAP2 binding sites (ii) Correspondence with CRISPR-based disruption of transcription factor binding sites
3. MalariaSED is further applied to reveal important aspects of Plasmodium biology. In particular the tool reveals that (i) non-coding variants that are strongly geographical differentiated show significantly higher predicted alteration in chromatin profiles relative to those with low geographical differentiation. These results are consistent with such non-coding variants underlying adaptation. (ii) non-coding variants in eQTLs for Artemisinin resistance, show higher predicted alteration in chromatin profiles than non-eQTL related SNPs. This suggests involvement of non-coding variants in eQTL response. (iii) non-coding variants observed in Plasmodium single cell data show strong evidence for selection, and have significant predicted impacts on transcription factor binding and epigenome.

While I enjoyed reading the paper, I have some concerns about the interpretation.

An exciting association is that highly differentiated non-coding variants show elevated predicted impacts on epigenetics, suggesting positive selection. However, an alternative explanation could be that these are deleterious changes that have spread to high frequency due to hitchhiking with selected coding variants. For example finding, such non-coding mutations close to drug resistance genes (or ART-R eQTLs) could indicate that these non-coding variants contribute to the resistance phenotype, but may simply indicate that they are mildly deleterious passengers.

This approach also provides opportunities for investigating other aspects of *P. falciparum* sequence variation. For example, conserved non-coding regions might be predicted to play an important regulatory function. Does MalariaSED predict that these regions impact chromatin binding or TF binding to, and does introduction of mutations within these regions change chromatin/TF binding.

The manuscript is mostly clearly and succinctly written. However, several sentences were hard to read and interpret, and could be rephased:

p7, 16-21. "An exceptional case for MalariaSED prediction occurred at the promoter region of the *pfmsp5* gene, where ChIP qPCR showed PfAP2-I binding but was not identified by the original ChIP-Seq signal⁵. Since MalariaSED was trained by the ChIP-Seq results, it only predicted a PfAP2-I binding probability lower than 0.1 for both the WT and ATGCA motif mutations." This is rather unclear - please rephrase and clarify

p8, line 59 "Our initial analysis of ~1.3 million noncoding variants⁸ 190 (trimming off ~1.4 noncoding loci with low..." ~1.4 [million??] non-coding loci

line 289. "One explanation is that MalariaSED may use DNA patterns where nucleotides are intermittently arranged for AP2-G5 binding predictions" I was not sure how to interpret this sentence - please rephrase

OVERALL - this is an exciting and novel direction in malaria biology that opens up many new opportunities. I look forward to seeing this approach applied to further aspects of Plasmodium biology and to see future epigenetic training datasets used to improve the power of this approach.

Reviewer 2

Were you able to assess all statistics in the manuscript, including the appropriateness of statistical tests used? No.

Were you able to directly test the methods? No.

Comments to author:

This manuscript title MalariaSED: a deep learning framework to decipher the regulatory contributions of noncoding variants in malaria parasites by Wang and colleagues describes a deep learning model that aims to predict various types of epigenetic signals from DNA sequence in several species of malaria parasites. Specifically, the model predicts various types of ChIP-seq histone modification signals, as well as ATAC-seq and ChIP-seq binding of AP2 and BDP1. The model achieves high classification accuracy, and the paper then proceeds to demonstrate how the model can be used to identify damaging genetic mutations, understand the effect of SNVs on TF binding, explore the interplay between variant effects and geography, and identify drug-related pathways. Overall, I found this manuscript hard to read and comprehend and would need substantial re-shaping to be accessible to a broad audience. Furthermore, some of the claims will need to be further validated using alternative approaches.

Major comments:

I found the introductory section of the paper and the initial description of the model and how it was trained and tested to be hard to follow and lacking in some very important details.

For example, in the introduction, the authors cite four existing models that predict epigenetic signals from DNA sequence but don't really discuss them. Indeed, I found it odd, in lines 56-64, that the authors spend so much time telling us about mechanistic and genetic differences between Plasmodium and other genomes. A priori, I don't think anyone would expect that a model trained exclusively in one species would likely generalize well to a different species, and certainly not when generalizing between species as distantly related as human and Plasmodium. The reason existing models are "not suitable for malaria parasites" is simply that they were not trained using Plasmodium data. The more important and unanswered question is whether the modeling approaches developed previously need to be modified to be applicable to Plasmodium data. My guess is that the answer to this question is "no." Hence, what is missing is a description of how the modeling choices in the four previously published methods differ from (and presumably build upon) one another, and how those influenced the author's choices when designing the proposed model. The authors should have taken one or more of those existing modeling approaches and simply trained that model using Plasmodium data. Why did they not do that? If the authors think that their proposed architecture is better than previously proposed architectures, then I think they need to demonstrate, with an experimental comparison, that their model outperform at least one of them.

At line 66, the authors say that they are doing "noncoding variant effect prediction," but at this point I still didn't actually know what they are predicting. This is a huge problem for understanding the paper. The abstract just says they are predicting "chromatin profiles," but that's a vague term. Remarkably, the first section of Results never clarifies this point. One can perhaps infer from the labels on the ROC curves in Figure 1 what is being predicted.

The claim, on line 94, that the LSTM is "crucial" is stated as fact but is a hypothesis. If the authors want to support this hypothesis, they will need to provide evidence in the form of an ablation experiment.

I can infer from the fact that they are using ROC curves that they must be predicting a binarized version of each datatype. This is a critical point, and one that should be mentioned and discussed in the main text. How, and more importantly, why do they threshold the observed values? Also missing is any mention of the loss function employed in the model. I found in the methods that they authors are using "weighted binary cross-entropy," but I don't know what kind of weighting was used.

Also missing from the first section is any description of how the data was split for training, validation, and testing. In perusing the Methods section, I discovered a potentially serious problem with the proposed approach, namely, that the data is based on a random split of 200bp windows. This is problematic because, as stated on line 90, the features include 1 kb of context around each window. The result is that if two adjacent windows get sent to the train and test sets, then the training set will contain features from the test set. This is exactly the mistake that Katie

Pollard's group made in training a promoter-enhancer predictor, which led to this critique:
<https://www.nature.com/articles/s41588-019-0434-7>

Overall, the authors use arbitrary values for most model parameters (window lengths, overlapping %, and so on). Variation in these values could affect performance dramatically. I think giving some sort of explanation as to how those values were chosen would be welcome.

The description of the model is incomplete, in the sense that we are not told how many filters are in each convolutional layer, the size of those filters, what the regularization parameters are, etc. There is a brief mention of hyperparameter optimization, but I wanted to know specifically what hyperparameters were optimized and over what ranges and using what objective function.

I would also add that samples size might be too small for obtaining any performance gains from using deep learning methods, in general. I think adding results for simpler classification methods for comparison would give a better baseline and some insights on the actual validity of the approach.

I might be missing something here, but the authors claim to be using the whole genome (including coding regions) as the dataset, split into 50,000 200bp bins. That amounts to 10Mb, but the genome itself is more than 20Mb? Does this discrepancy need an explanation?

Minor comment:

I think the discussion is weak. It seems just a recapitulation of the results.

Dear Editor

We would like to thank you for giving us the opportunity to further improve our study. We appreciate the positive and constructive comments and suggestions on our manuscript (Manuscript ID: GBIO-D-22-00875) made by the editors and the reviewers. The comments are valuable and helpful for revising and improving our paper, and will serve as a guide to our future research. We have carried out further analysis in line with the reviewers' suggestions and modified our manuscript accordingly.

The major concerns brought to our attention by Reviewer 1 included the potential deleterious changes resulting from high-frequency non-coding variants, along with the contribution of conserved non-coding regions to regulatory activity. We have systematically investigated the relationship between evolution conservation and regulatory activity, as well as the regulatory activity change introduced by potential mutations. We also used our previously published parasite growth phenotype of large-scale *piggyBac* mutants to interrogate the possibility of exploring the deleterious effects of non-coding variants. We thank the encouraging words from Reviewer 1. This motivates us to develop new prediction models in Apicomplexan research field.

The major concerns brought by Reviewer 2 about the model performance justification, including comparison with previously published models in the mammalian genomes, model training in the separated chromosome, the contribution of the LSTM layer, along with performance comparisons with simple classification methods, were also addressed in our revision. We also thank the comments from Reviewer 2 about the necessity to strength model description. This helps us to improve the clarity of our manuscript.

We sincerely hope that you find that the new data and the manuscript revision have improved the quality of the paper in a significant manner. Major modifications have been marked in color in the revised manuscript, and below is a point-by-point response to the reviewers' comments and suggestions

Best wishes

Chengqi Wang, PhD

Reviewer #1: This is a thought provoking paper that uses a deep learning approach to predict the phenotypic impact of non-coding mutations. This is done by training a model, using existing experimental epigenetic data for Plasmodium. The approach is of specific interest to those working on malaria parasites and of broad general interest to those working with a wide range of organisms.

The paper has several strong features:

1. This is extremely important topic, given that there are quite similar numbers of coding and non-coding variants in the Plasmodium genome, and only coding variants have receive significant attention. Furthermore, existing models for making similar predictions in other model organisms work poorly in Plasmodium, most likely because *P. falciparum* is extremely AT biased, and markedly different from most other organisms in having rather few transcription factors.

2. The development and validation of the malariaSED tool is carefully done, with careful optimization of the size of DNA windows for maximal performance, and evaluation of predictive power by showing: (i) Accurate prediction of ApiAP2 binding sites (ii) Correspondence with CRISPR-based disruption of transcription factor binding sites

3. MalariaSED is further applied to reveal important aspects of Plasmodium biology. In particular the tool reveals that (i) non-coding variants that are strongly geographical differentiated show significantly higher predicted alteration in chromatin profiles relative to those with low geographical differentiation. These results are consistent with such non-coding variants underlying adaptation. (ii) non-coding variants in eQTLs for Artemisinin resistance, show higher predicted alteration in chromatin profiles than non-eQTL related SNPs. This suggests involvement of non-coding variants in eQTL response. (iii) non-coding variants observed in Plasmodium single cell data show strong evidence for selection, and have significant predicted impacts on transcription factor binding and epigenome.

We sincerely appreciate the encouraging words from reviewer one!

While I enjoyed reading the paper, I have some concerns about the interpretation.

An exciting association is that highly differentiated non-coding variants show elevated predicted impacts on epigenetics, suggesting positive selection. However, an alternative explanation could be that these are deleterious changes that have spread to high frequency due to hitchhiking with selected coding variants. For example finding, such non-coding mutations close to drug resistance

genes (or ART-R eQTLs) could indicate that these non-coding variants contribute to the resistance phenotype, but may simply indicate that they are mildly deleterious passengers.

We appreciate Reviewer 1 bringing up this alternative explanation brought. This is important. We conducted an analysis to investigate whether geographically differentiated variants accompanied by different chromatin

profile effects in the noncoding region are more likely closer to genes with high levels of geographic differentiation. This should be one of the consequences of hitchhiking with selected coding variants. The analysis was based on the global Fst score calculated by MalariaGEN (MalariaGEN, Wellcome Open Research, 2021) for each gene based on the highest-ranking non-synonymous variants within a gene. Our results show geographically differentiated noncoding variants with higher alteration in AP2-G binding profile in commit schizonts, AP2-G5

in trophozoites, chromatin accessibility, and euchromatin tend closer to the gene harboring highly differentiated non-synonymous variants (Fig. A, supplement Figure 5 in the manuscript for all chromatin profiles). The results imply that the non-synonymous variants with high geographic differentiation in the malaria parasite could hitchhike neighboring noncoding variants to enhance or repress the gene expression by altering the chromatin compactness.

We next try to answer another question brought up by reviewer 1. Whether the high-frequency non-coding variants hitchhiked by selected coding variants present mildly deleterious? We used the growth phenotype during *P. falciparum* asexual development of > 27,500 non-coding *piggyBac* transposon insertion sites (Zhang et al., Science, 2018) to question whether significant regulatory activity change brought by *piggyBac* insertion is associated with parasite growth rates. Nonetheless, the

~3kb *piggyBac* insertion brings several uncertainties in estimating the regulatory effects to the neighboring gene. 1) The ~3kb *piggyBac* insertion may directly bring in regulatory profile changes either in 5' or 3', as well as regulatory changes on the transfected plasmids after transfection into the parasite genome (Fig. B); 2) The *piggyBac* insertion may impact the original regulatory relationship between a non-coding functional element and its target gene due to the altered distance brought by ~3kb plasmid. 3) Non-coding mutants lethal to parasite growth were not able to recover. Although we controlled the chromatin effects on both 5' and 3', requiring both sides have the same direction of chromatin effects (chromatin effects on both 5' and 3' should be both positive and negative changes compared with wild-type, predicted by MalariaSED), we did not observe a significant growth difference (measured by QIseq reads counts, Zhang et al., Science 2018) when compared with the remaining mutants for all 13 chromatin profiles. Therefore, current data resources in the malaria research community cannot either support answering this question or develop a model to predict the deleterious effects of non-coding variants in parasites.

This approach also provides opportunities for investigating other aspects of *P. falciparum* sequence variation. For example, conserved non-coding regions might be predicted to play an important regulatory function. Does MalariaSED predict that these regions impact chromatin binding or TF binding to, and does introduction of mutations within these regions change chromatin/TF binding.

Great point! We conducted the analysis to answer the two questions from Review 1: 1) whether the conserved synteny regions (DeBarry et al., Molecular Biology and Evolution, 2011) tend to have different chromatin profiles compared with other

genome regions. 2) whether these synteny regions in *P. falciparum* are more tolerant to genetic variants predicted by MalariaSED? Using published chromatin profiles, also utilized for MalariaSED development, we observed significantly higher open chromatin and AP2-G5 binding profiles on the synteny region (Fig.C, Wilcoxon test $< 1e-3$). To answer the second question, we randomly introduced a particular number of single nucleotide substitutions (SNSs) to synteny regions. The synteny regions harboring SNSs were enlarged to 1kb and predicted the chromatin effects by MalariaSED. The variant introduction step was repeated 100 times for each synteny region. We only kept noncoding synteny regions shorter than 1kb for analysis since MalariaSED can only predict genomic regions with a 1kb length. The length of the majority of synteny regions (99%) reported previously is shorter than 633bp, and ~85% of them shorter than 100bp. This guaranty most of the synteny regions were involved in the analysis. As shown in Fig. D, we observed that the introduction of SNSs on the synteny region is significantly likely to alter open chromatin profiles in parasite rings (Wilcoxon test $p < 2.2e-16$ compared with random genome regions). Other profiles include AP2-G5 in *P. falciparum* trophozoites, pfBDP1 in schizonts and open chromatin in late rings and schizonts (Wilcoxon test $p < 1e-3$ compared with random genome regions).

The manuscript is mostly clearly and succinctly written. However, several sentences were hard to read and interpret, and could be rephased:

p7, 16-21. "An exceptional case for MalariaSED prediction occurred at the promoter region of the *pfmsp5* gene, where ChIP qPCR showed PfAP2-I binding but was not identified by the original ChIP-Seq signal⁵. Since MalariaSED was trained by the ChIP-Seq results, it only predicted a PfAP2-I binding probability lower than 0.1 for both the WT and ATGCA motif mutations." This is rather unclear - please rephrase and clarify

Thanks to the reviewer for pointing out this ambiguous sentence. We have modified it to clarify the meaning.

p8, line 59 "Our initial analysis of ~1.3 million noncoding variants⁸ 190 (trimming off ~1.4 noncoding loci with low..." ~1.4 [million??] non-coding loci

We apologize for this mistake. Yes, it should be 1.4 million.

line 289. "One explanation is that MalariaSED may use DNA patterns where nucleotides are intermittently arranged for AP2-G5 binding predictions" I was not sure how to interpret this sentence - please rephrase

We apologize for this unclear description. It should be the multiple DNA sequence patterns far apart from each other in *P. falciparum* genome. We have revised this sentence.

Reviewer #2: This manuscript title MalariaSED: a deep learning framework to decipher the regulatory contributions of noncoding variants in malaria parasites by Wang and colleagues describes a deep learning model that aims to predict various types of epigenetic signals from DNA sequence in several species of malaria parasites. Specifically, the model predicts various types of ChIP-seq histone modification signals, as well as ATAC-seq and ChIP-seq binding of AP2 and BDP1. The model achieves high classification accuracy, and the paper then proceeds to demonstrate how the model can be used to identify damaging genetic mutations, understand the effect of SNVs on TF binding, explore the interplay between variant effects and geography, and identify drug-related pathways.

We thank the reviewer for the comprehensive summary of our work and contribution.

Overall, I found this manuscript hard to read and comprehend and would need substantial re-shaping to be accessible to a broad audience. Furthermore, some of the claims will need to be further validated using alternative approaches.

We thank the reviewer's comments and deeply feel the necessity and importance of justifying the outperformance of MalariaSED. Please check the following point-by-point response to the reviewers' comments and suggestions.

Major comments:

I found the introductory section of the paper and the initial description of the model and how it was trained and tested to be hard to follow and lacking in some very important details.

We agreed with the reviewer's comments and provided all details of model development in our revision. We apologize for not providing enough description in our original manuscript, and we have made substantial revisions to demonstrate the outperformance of MalariaSED in epigenetic marker predictions in the malaria parasites. We first emphasize that MalariaSED predicts the probability of presenting the signal peak of different chromatin profiles in both the introduction and the beginning of the result. Secondly, we list all optimized hyperparameters, including the searching range in Bayes optimization, in supplement table 1. Thirdly, we added more words to describe the model training process in the method part. At last, we uploaded all training scripts to GitHub.

For example, in the introduction, the authors cite four existing models that predict epigenetic signals from DNA sequence but don't really discuss them. Indeed, I found it odd, in lines 56-64, that the authors spend so much time telling us about mechanistic and genetic differences

between Plasmodium and other genomes. A priori, I don't think anyone would expect that a model trained exclusively in one species would likely generalize well to a different species, and certainly not when generalizing between species as distantly related as human and Plasmodium. The reason existing models are "not suitable for malaria parasites" is simply that they were not trained using Plasmodium data. The more important and unanswered question is whether the modeling approaches developed previously need to be modified to be applicable to Plasmodium data. My guess is that the answer to this question is "no." Hence, what is missing is a description of how the modeling choices in the four previously published methods differ from (and presumably build upon) one another, and how those influenced the author's choices when designing the proposed model. The authors should have taken one or more of those existing modeling approaches and simply trained that model using Plasmodium data. Why did they not do that? If the authors think that their proposed architecture is better than previously proposed architectures, then I think they need to demonstrate, with an experimental comparison, that their model outperform at least one of them.

We appreciate that the reviewer wrote the paragraph with rigid logical sentences suggesting we provide the rationale of modeling choices. In our revision, we have conducted a series of computational analyses to demonstrate the outperformance of MalariaSED in the malaria parasites to the existing state-of-the-art DL models.

The two methods mentioned in the (Zirmec et al., Nat Commun, 2020; Zhou et al., Nat Genet, 2018; ref 22 and 24 in the manuscript) are DL models mainly based on 3-layer convolutional network. We therefore applied three convolutional layers with the same kernel number as Zhou's method (Zhou et al., Nat Genet, 2018). Since the third convolutional layer is the replacement of the LSTM layer in MalariaSED, the performance of the convolutional network would also evaluate the necessity of the LSTM layer. The supplement table 4 lists all hyperparameters and their searching range in Bayes optimization. We used the same training/validation/test set as MalariaSED for the performance comparisons. The comparisons indicate MalariaSED with LSTM layer achieved better performance in most chromatin profile predictions than the three-layer convolutional network (Average auROC 0.96 vs. 0.93, auPRC 0.73 vs. 0.71 in test set, Supplement Table 2). The only two exceptional cases were PbAP2-O and PfAP2-G in sexual rings.

The other methods mentioned in the introduction (Kelley et al., Genome Res, 2018; Avsec et al., Nat Methods, 2021, Ref 21 and 23 in the manuscript) are Basenji2 and Enformer. Enformer was improved from Basenji2. Basenji2 is the former state-of-the-art chromatin profile predictors from DNA sequence. The transformer mechanism has shown the capacity to capture distal elements in a sequence. Enformer embraces this idea and replaces the dilated convolution layers of Basenji2 with transformer layers to increase the receptive field of the model. Considering the success of Enformer in incorporating chromatin profiles in a large range of genome scales, we tested the input sequence with different lengths (12.8, 32 and 64 kb) covering all 13 chromatin profiles in *P. falciparum* genome for MalariaSED development. The Enformer finally outputs 50, 125, and 250 binned predictions so that each bin (sequence position vector) represents 128bp (Supplement table 5). We finally select the input sequence length of 64 kb for the comparison with MalariaSED, since Enformer with this input sequence length achieved the highest Pearson correlation, 0.63 globally in *P. falciparum* genome. The final output from Enformer is 250 (each position represents 128bp bin) x 13 (chromatin profiles). We assigned each output bin (128bp) from Enformer a label based on whether the output is within a reported epigenetic peak, which is the same as MalariaSED. The auROC and auPRC were calculated based on the value output from Enformer and the assigned label. The average auROC and auPRC from Enformer only obtain 0.65 and 0.1 for peak identification of different chromatin profiles. We also systematically discussed the reason behind the low performance of Enformer in identifying chromatin profile peaks. This may be owing to the peaks reported in

several publications using overlapped regions of multiple replicates (Santos et al., Cell Host Microbe, 2017; Josling et al., Nat Commun, 2020; Josling et al., Cell Host Microbe, 2015; Ref 5-7 in the manuscript), or the use of peak calling results from only one biological replicate (Toenhake et al., Cell Host Microbe, 2018, Ref 29 in the manuscript), which leads to the deviation between peak location and sequencing signal averaged from different replicates. Enformer is a powerful tool to predict different chromatin profiles. It is originally developed in the mouse and human genomes and has demonstrated a promising application in Apicomplexa parasites. Nonetheless, the current tools for epigenetic profile peak calling are based on sequencing read counts (review article, Thomas et al., Brief Bioinform, 2017, Ref 42 in the manuscript), which may not be suitable for handling Enformer prediction results. Further efforts need to be conducted to investigate whether Enformer can be directly utilized to train on reported peaks of chromatin profiles.

At line 66, the authors say that they are doing "noncoding variant effect prediction," but at this point I still didn't actually know what they are predicting. This is a huge problem for understanding the paper. The abstract just says they are predicting "chromatin profiles," but that's a vague term. Remarkably, the first section of Results never clarifies this point. One can perhaps infer from the labels on the ROC curves in Figure 1 what is being predicted.

We apologize for not presenting a clear definition of MalariaSED prediction. In our revision, we have clearly described the prediction from MalariaSED as the probability of presenting signal peaks of different chromatin profiles within the input sequences. The sequence-based feature learning enables MalariaSED to predict epigenetic effects based upon sequence alterations at single-nucleotide resolution.

The claim, on line 94, that the LSTM is "crucial" is stated as fact but is a hypothesis. If the authors want to support this hypothesis, they will need to provide evidence in the form of an ablation experiment.

Great point! In our revision, we have performed ablation analyses to change the last layer, LSTM layer in MalariaSED, to a convolutional layer. The three-convolutional layer with the same kernel number as Zhou's method (Zhou et al., Nat Genet, 2018) was trained exactly as MalariaSED, including hyperparameter optimization and train/validation/test set. We compared the performance of this convolutional network with MalariaSED, showing the results in Supplement Table 2. The results demonstrate the outperformance of MalariaSED on the three-layer convolutional network, which is the model structure of Zhou's method.

I can infer from the fact that they are using ROC curves that they must be predicting a binarized version of each datatype. This is a critical point, and one that should be mentioned and discussed in the main text. How, and more importantly, why do they threshold the observed values? Also missing is any mention of the loss function employed in the model. I found in the methods that they authors are using "weighted binary cross-entropy," but I don't know what kind of weighting was used.

Thanks to the reviewer for raising this question. We divided the whole parasite genome into bins of 200 bp, yielding more than 50,000 bins not overlapping with the gene body. A bin is labeled "positive" if more than 60% of it overlaps with ChIP or ATAC peaks, while the remaining bins are used as negative datasets. The ROC and PR curves were generated based on MalariaSED

prediction from the sigmoid activation function and the label assigned for each 200bp bin. We have mentioned the generation of ROC and PR curves at the method in our revision. We apologize for the misleading term 'weighted binary cross-entropy' in the manuscript. The weight we used is 1:1, and it should be just binary cross-entropy. MalariaSED was trained independently in 15 chromatin profiles from *P. falciparum* and *P. berghei*. MalariaSED was implemented in TensorFlow (v2.4.0), and was trained on 4 GPU Tesla V100 with batch size 2,000. The best model was determined by the highest auPRC in the validation set and evaluated by the test set.

Also missing from the first section is any description of how the data was split for training, validation, and testing. In perusing the Methods section, I discovered a potentially serious problem with the proposed approach, namely, that the data is based on a random split of 200bp windows. This is problematic because, as stated on line 90, the features include 1 kb of context around each window. The result is that if two adjacent windows get sent to the train and test sets, then the training set will contain features from the test set. This is exactly the mistake that Katie Pollard's group made in training a promoter-enhancer predictor, which led to this critique: <https://www.nature.com/articles/s41588-019-0434-7>

We appreciate this important comment raised by the reviewer. While we admit that partially overlapped sequences between training/validation/test sets can inflate the model performance, the extremely low number of some chromatin profile peaks in *P. falciparum* (AP2I, AP2G and PfBDP1) may lead some chromosomes to have low numbers or even don't have any peaks (e.g., Chromosomes 1 and 9 don't have PfBDP1 peaks in trophozoites). While split by chromosomes can effectively separate unwanted information sharing, it misses the opportunity to study the effects in those left-out chromosomes. Our use of random split is a trade-off for less bias over less variance.

To examine the model's generalizability in completely unseen data, we retrain the model in a chromosome-split manner. To break the dependence between samples in training, validation, and test sets, we split samples such that all samples on the same chromosome were all in one of the training, validation and test set. We select chromosome combinations in the training, validation, and test set so that the ratio is close to 7/1.5/1.5 for the chromatin profiles with low positive samples (positive bin number < 1,100). For the remaining chromatin profiles, bins on chromosomes 10 and 11 were selected as the validation set and chromosomes 9 and 12 as the test set. The bins in all other chromosomes were used as a training set (Supplement table 3). We stuck to the training process of the original MalariaSED. The results show an average 0.03 decreased performance in auROC (0.96 vs. 0.93) and 0.14 decrease in auPRC (0.73 vs. 0.59, the former indicates random splitting) compared with the random splitting data set (Supplement table 3 in the manuscript). The results indicate the necessity to use chromosome-splitting strategy to evaluate model performance. We have added the performance of MalariaSED developed in the chromosome-split strategy in the first paragraph of the results in the manuscript and discussed it in the manuscript.

Overall, the authors use arbitrary values for most model parameters (window lengths, overlapping %, and so on). Variation in these values could affect performance dramatically. I think giving some sort of explanation as to how those values were chosen would be welcome.

We agreed with the reviewer's comment that the window length and surrounding region would impact model performance. We have developed a DL model to enhance the prediction performance for each of the 15 chromatin profiles in the malaria parasites. The 200bp region and its surrounding 1 kb sequence context have been used for DL model development in the human genome (Zhou et al., Nat Methods, 2015; Chen et al., BMC Bioinformatics, 2021, ref 32 and 47 in the manuscript). We have added these two references in our revision, see line 495. We also presented the performance for selecting the different lengths of surrounding windows in Supplement figure 1. The revision also listed all the hyperparameters and their searching range in Bayes optimization.

The description of the model is incomplete, in the sense that we are not told how many filters are in each convolutional layer, the size of those filters, what the regularization parameters are, etc. There is a brief mention of hyperparameter optimization, but I wanted to know specifically what hyperparameters were optimized and over what ranges and using what objective function.

We thank the reviewer for pointing out this. In our revision, we have improved the description of each hyperparameter so that it is consistent with the terminology used in the TensorFlow package. In supplement tables 1 and 4, we provided all hyperparameter names, description and their searching range in Bayes optimization. The objective of Bayes optimization used in hyperparameter optimization is the maximum of auPRC. The size of each filters and regulation parameters have been provided as 'kmer' and 'convL2'.

I would also add that samples size might be too small for obtaining any performance gains from using deep learning methods, in general. I think adding results for simpler classification methods

for comparison would give a better baseline and some insights on the actual validity of the approach.

To examine whether the simpler classification model could provide a similar performance as MalariaSED, we trained 200bp ATAC-Seq peak by using the SVM model from the idea in the human genome (Lee et al., Genome Res, 2011, PMID: 21875935). We used the k-mer (oligomers of length k) to extract surrounding 1kb sequence information as MalariaSED. To identify the

k-mer enrichment in open chromatin region, 20-fold more intergenic regions excluding known open chromatin regions, are extracted as negative sequences. We generated these set to match the distribution of sequence length, repeat element fraction (each k-mer sequence pattern frequency) and GC content to the corresponding positive set (Fig. E). The frequency of 7-, 8-, 9- and 10-mer were used to generate the original feature set. Instead of counting only an

exact k-mer, its reverse complement is also counted, and then redundant k-mers are removed. The training/validation/test set was randomly generated to follow the ratio 7/1.5/1.5. The top 1,000 K-mer features are selected by their relative frequency in the positive set based on Fisher's exact test (Li et al., Nucleic Acids Research, 2017, PMID: 27980060). The SVM with RBF kernel was used to train on the training set and the best model was selected based on the highest auPRC in the validation set. Table A lists the performance of the selected SVM model in

	AUC_ROC	AUC_PR
ATAC_Seq_05_10h	0.56	0.02
ATAC_Seq_15_20h	0.55	0.03
ATAC_Seq_25_30h	0.57	0.09
ATAC_Seq_35_40h	0.52	0.15

Table A. The performance of SVM on the test set

the test set. It shows a very limited performance of SVM, when traditional machine learning methods and hand-craft features were adopted, in discriminating ATAC-Seq profiles.

I might be missing

something here, but the authors claim to be using the whole genome (including coding regions) as the dataset, split into 50,000 200bp bins. That amounts to 10Mb, but the genome itself is more than 20Mb? Does this discrepancy need an explanation?

We thank the reviewer for pointing out this important issue. We only include 200bp bins not overlapped with the coding regions for MalariaSED development, which resulted in only half of the genome was processed. We have added this to the manuscript.

Minor comment:

I think the discussion is weak. It seems just a recapitulation of the results.

We thank the reviewer's comments, which represent the concern of a group audience, especially in the field of algorithm development. In our original manuscript, we mainly discussed the observation or results from the application of MalariaSED in predicting different genomic variants in the malaria parasites. Some results from a biological perspective may have multiple explanations that need to be an emphasis. We apologize for not providing sufficient discussion for the model evaluations and comparisons, and we truly appreciate the valuable comments provided by Reviewer 2 regarding model performance justification and model description. In our revision, we systematically discussed the performance comparisons with existing state-of-the-art DL models developed in model organisms, MalariaSED evaluation in chromosome splitting manner, and the possible explanation of the outperformance of MalariaSED. We sincerely hope that the new data and the manuscript revision have improved the quality of the paper.

Second round of review

Reviewer 1

The reviewers have addressed my queries regarding the interpretation of the results from MalariaSED, and possible applications to interesting aspects of Plasmodium genomics. They have also clarified the text where requested.

However, reviewer 2 raised a number of important technical questions, which clearly need addressing. I leave Rev 2 to address whether the technical critiques have been addressed, as this is not my expertise. However, I found that the added text addressing these issues was quite dense and hard to read. For example, some sentences in the discussion need rewriting to improve clarity: e.g. "The comparison with this convolutional network also demonstrated the contribution of the LSTM layer to prediction performance since the last LSTM layer in MalariaSED was replaced by a convolution layer." I have also added some suggested rephrasing in minor comments.

Additional editing for clarity would greatly improve this manuscript (and particularly the new sections) accessible to a broader audience.

Minor points

Pg 12, lines 45-8 "Since the majority of prediction frameworks in chromatin prediction were developed in model organisms²¹⁻²⁴, we therefore interrogate whether these frameworks can be applied in the malaria parasites and comparable with MalariaSED".

Change to "....and IS comparable with MalariaSED"

Pg 13, lines 16-17: "The only two exceptional cases were PbAP2-O and PfAP2-G in sexual rings."

Should this read "sexual stages" or gametocytes?

Pg19, 39-44 "We stuck to the training process of MalariaSED on random splitting data set."

Replace by: We retained the same training process for both the random splitting and chromosome splitting data set.

Pg21, lines 26-30 "It could be mildly deleterious noncoding variants contributing to the resistance or other potential benefit phenotypes to parasite growth. This requires a large-scale genetic screen to be undertaken to investigate reported noncoding variants in the parasite's normal growth condition."

This new text was unclear to me and needs rephrasing

Reviewer 3

The authors have addressed some of the comments made in the initial review. A few still remain, including ones based on information added in the revision. One general observation about the general tone of the manuscript, is that the authors try, somewhat unsuccessfully, to demonstrate the advantage of the chosen architecture. I was not convinced by those arguments. Instead, I suggest to focus on the strengths of the work, which is an application of deep learning to an interesting organism with an unusual genome.

Comments:

1. The initial review raised the following point:

Also missing from the first section is any description of how the data was split for training, validation, and testing. In perusing the Methods section, I discovered a potentially serious problem with the proposed approach, namely, that the data is based on a random split of 200bp windows. This is problematic because, as stated on line 90, the features include 1 kb of context around each window. The result is that if two adjacent windows get sent to the train and test sets, then the training set will contain features from the test set. This is exactly the mistake that Katie Pollard's group made in training a promoter-enhancer predictor, which led to this critique: <https://www.nature.com/articles/s41588-019-0434-7>

I found the authors' response unsatisfactory. There are two ways to approach this point:

1. Perform testing on left-out chromosomes, which is the most robust way of performing this kind of analysis.
2. Perform an analysis of the potential overlap between peaks. The authors state that this is likely to be low due to the sparsity of ChIP-seq peaks in *P. falciparum*, but fail to provide quantitative data to support this claim. Instead they state that "While split by chromosomes can effectively separate unwanted information sharing, it misses the opportunity to study the effects in those left-out chromosomes. Our use of random split is a trade-off for less bias over less variance." There are two false claims here. First, if they really want to provide test results on the entire genome, they can perform a cross-validation analysis. Even with their current analysis they are missing lots of peak regions from testing. Second, the term "bias" here is mis-used and demonstrates a lack of understanding of the bias/variance tradeoff in machine learning. Infact, their results on left out chromosomes as suggested by the reviewer, suggest that this is indeed an issue.

2. In the introduction the authors note that

"Sequence-based prediction models provide a new perspective on the effects of genomic variants on gene regulation. However, the majority of available models were developed in model organisms, which are not suitable for malaria parasites due to several unusual features of their genome architecture"

As mentioned in the initial review, this statement is inaccurate. The authors use existing model architectures very similar to those used in model organisms, contradicting what they have said. I suggest rewording as follows:

"Sequence-based prediction models provide a new perspective on the effects of genomic variants on gene regulation. However, the majority of available models were developed in model organisms, and their applicability to malaria parasites whose genomes have several unusual features, requires careful evaluation."

In this context the authors note that "the applicability of existing predictive models include the apparent absence of some subunits of the preinitiation complex". I don't see how this is case - please explain how that changes anything about the modeling approach.

3. "This framework is complementary to experimental approaches at higher throughput with lower cost". While it's true that it is complementary to experimental approaches, it's not meant to

replace them - you need the experimental data to be able to perform this type of analysis. Rather, your framework enables to make the most out of the existing experimental data and derive biological insight from it (e.g. effect of non-coding variants).

4. "The unusual genome property (e.g., ~80% AT content on average, rising to ~92% in intergenic regions) of malaria parasites requires the development of a novel computational framework to understand the regulatory sequence code from parasite noncoding sequences."

The proposed framework is not novel; I suggest rephrasing as follows:

The unusually high AT content of the *P. falciparum* genome (80% AT on average, rising to ~92% in intergenic

regions) requires careful evaluation of existing deep learning approaches for understanding the regulatory sequence code from parasite noncoding sequences.

5. "Here, s is an input sequence represented as a $4 \times l$ matrix, where each row represents a nucleotide, and each column indicates a position in the length l sequence. '1' represents a specific nucleotide present at a specific position of the sequence, and '0' does not (Fig. 1A)."

Suggest replacing with:

"Here, s is an input sequence represented as a $4 \times l$ matrix, where each column represents a position in the sequence with a "1" in the row corresponding to the nucleotide at that position, and has zeros otherwise."

6. "The convolution layer (Conv) is a motif scanner analogizing the position weight matrix." I think you meant that

The convolution layer (Conv) functions as a motif scanner analogous to a position weight matrix.

7. "The LSTM is crucial since TF binding sites may have a group of sequence patterns separated from each other by a long distance" You provide justification for this statement later, but should be referred to here as well.

8. The new section "MalariaSED prediction in malaria parasites outperforms state-of-the-art frameworks"

developed in model organisms" is poorly conceived. The good part is that it demonstrates the advantage of using the LSTM layer. However, I don't think that the ENFORMER architecture comparison makes a valid statement. The ENFORMER framework is designed for much larger datasets such as those that one finds in large genomes like the human genome. Therefore, I do not find it surprising that it did not perform very well, and don't see the point of reporting on such a failed experiment. Here you are opening yourselves to the criticism that perhaps it didn't work because you did not optimally select its hyperparameters. The contribution of your work is on demonstrating the ability of DL to work well in this interesting genome, using commonly used DL techniques. I suggest simply reporting the fact that LSTM provided improved performance over a purely convolutional approach in your experiments as part of the previous section where the statement about the contribution of the LSTM to performance was made. I would also like to note that the purely convolutional approach you compare to was state-of-the-art in 2015 which is ages ago when it comes to developments in DL. Therefore the statement that MalariaSED outperforms state of the art frameworks is not really valid.

9. "We also tested the MalariaSED performance on chromosome splitting strategy with average auROC = 0.93 and auPRC = 0.59 (Supplement Table 3)."

This statement does not correctly represent the fact that this is the preferred strategy. Please rephrase e.g. as:

We also tested the MalariaSED performance on a chromosome splitting strategy with chromosomes XX as the test set, achieving an average auROC = 0.93 and auPRC = 0.59 (Supplement Table 3). We note that this is the preferred evaluation strategy as it avoids biases that result from potentially overlapping examples (see e.g. <https://www.nature.com/articles/s41576-021-00434-9>).

This educates the readers on the proper way of running their experiments, and demonstrates that it does have an effect.

10. In the first section of the Results section I would like to see a bit more information about what you are predicting:

There is a single sentence stating that you are using "large-scale epigenetic experimental data, including open chromatin accessibility (ATAC-Seq)²⁹, TF binding, and histone-mark profile (ChIP-Seq)" I would have liked to see more information about how many overall profiles are being predicted (the number appears later) and which TFs and histone-marks are being profiled. As is, it is lacking concreteness.

11. I would have liked to see that your model is able to learn binding motifs that are similar to the known motifs of ApiAP2s transcription factors.

12. "Single-nucleotide substitutions at the TF motif weaken TF binding" This is exactly what we would expect, since the convolutional layer is a collection of motif detectors! This is a standard observation in studies that use DL for genomics data.

13. The previous review noted that the Discussion re-iterates many points introduced in the Results section and do not add to the manuscript. The discussion needs to contain NEW insights regarding the results rather than a rehashing of what you already wrote. Furthermore, my gripes with the discussion of your Enformer results hold even more for what you wrote in the Discussion.

14. In the purely convolutional network you used hyperparameters similar to those used to analyze the human genome. This is despite the fact that the malaria genome is much smaller, and that you modeled a dataset consisting of a lot fewer experimental datasets. So this model was not provided with the opportunity to do well, so obviously did not perform as well as the LSTM based version.

15. In your svm experiments you used k-mers that are very long. In fact, the original paper you cite mentions that "predictive power begins to decrease when k is greater than six. You used values of seven and above. A better SVM method to compare to is the GKM-SVM method that implements a more flexible kernel method.

16. The authors mention that their method was trained using histone-mark profile data, but no other information is provided in this matter - which histone-mark, where did the data come from, and what was its contribution to network performance.

17. Very few TFs are known in the malaria parasite as noted by the authors. This could be the result of poor annotations rather than an organism with very few TFs. Given that the network was trained on chromatin accessibility data, it might have picked up signals from TFs that are not represented in the TF- ChIP-seq datasets. Strong motifs that do not match to known ApiAP2 motifs are candidates for novel TFs. This could be a good avenue for further research.

Typos and grammar

"experimental data, including open chromatin accessibility (ATAC-Seq), TF binding, and histone-mark profile  profiles

"We use early stopping to seize the parameter optimization process if there is no improvement in validation auPRC for ten consecutive epochs." replace seize with "halt"

The reference "Practical Bayesian Optimization of Machine Learning Algorithms" was published in the Neurips conference. Also, the method itself is called

"with the objective to maximum auPRC"  objective to maximize

The ROC and PR curves were generated based on the MalariaSED prediction from the sigmoid activation function...  from the network output...

Authors Response

Point-by-point responses to the reviewers' comments:

Reviewer reports:

Reviewer #1: The reviewers have addressed my queries regarding the interpretation of the results from MalariaSED, and possible applications to interesting aspects of Plasmodium genomics. They have also clarified the text where requested.

We extend our heartfelt gratitude to Reviewer 1 for their valuable efforts and insightful questions, which have significantly contributed to the improvement of our manuscript.

However, reviewer 2 raised a number of important technical questions, which clearly need addressing. I leave Rev 2 to address whether the technical critiques have been addressed, as this is not my expertise. However, I found that the added text addressing these issues was quite dense and hard to read. For example, some sentences in the discussion need rewriting to improve clarity: e.g. "The comparison with this convolutional network also demonstrated the contribution of the LSTM layer to prediction performance since the last LSTM layer in MalariaSED was replaced by a convolution layer." I have also added some suggested rephrasing in minor comments.

Additional editing for clarity would greatly improve this manuscript (and particularly the new sections) accessible to a broader audience.

We appreciate the comments and feedback provided. In our latest revision, we have addressed the concern by toning down the comparisons with established DL models and focusing on the applications of the model in malaria genomes. Furthermore, the specific sentence mentioned has been removed to ensure clarity and ease of understanding for a broad audience interested in DL applications for improving and understanding parasite biology.

Minor points

Pg 12, lines 45-8 "Since the majority of prediction frameworks in chromatin prediction were developed in model organisms²¹⁻²⁴, we therefore interrogate whether these frameworks can be applied in the malaria parasites and comparable with MalariaSED".

Change to " ...and IS comparable with MalariaSED

This sentence has been removed in the revision as we have reduced the comparisons with existing DL frameworks developed in model organisms.

Pg 13, lines 16-17: "The only two exceptional cases were PbAP2-O and PfAP2-G in sexual rings."

Should this read "sexual stages" or gametocytes?

We appreciate the reviewer's comment, and in response, we have made the necessary modification to use the term "sexual stages" in the revised manuscript. (Page 5, line 36-37)

Pg19, 39-44 "We stuck to the training process of MalariaSED on random splitting data set."

Replace by: We retained the same training process for both the random splitting and chromosome splitting data set.

Following the suggestion of Reviewer 1, we have replaced the sentence pointed out with the one recommended by the reviewer in the revision (Page 15, line 40-41).

Pg21, lines 26-30 "It could be mildly deleterious noncoding variants contributing to the resistance or other potential benefit phenotypes to parasite growth. This requires a large-scale genetic screen to be undertaken to investigate reported noncoding variants in the parasite's normal growth condition."

This new text was unclear to me and needs rephrasing

We have rephrased to 'The subsequent step involves assessing the fitness cost of these noncoding variants. It is plausible that these variants may exert mild deleterious effects, contributing to resistance or other potential advantageous phenotypes for parasite growth. To explore this further, a large-scale genetic screen is required to investigate the reported noncoding variants under the parasite's normal growth conditions. Such investigations would provide valuable insights into the functional implications of these variants and their impact on the overall fitness of the parasite' (Page 13, line 50- Page 14, line13).

Reviewer #3:

The authors have addressed some of the comments made in the initial review. A few still remain, including ones based on information added in the revision. One general observation about the general tone of the manuscript, is that the authors try, somewhat unsuccessfully, to demonstrate the advantage of the chosen architecture. I was not convinced by those

arguments. Instead, I suggest to focus on the strengths of the work, which is an application of deep learning to an interesting organism with an unusual genome.

We extend our sincere appreciation to Reviewer 3 for acknowledging the dedication we have invested in applying DL models in malaria genome research. In response to the feedback from Reviewer 3, we have made the necessary adjustments to tone down the comparisons with established DL frameworks in model genomes. Additionally, We have revised the discussion section in our manuscript to focus on potential future applications that could further enhance our model's performance and capabilities. Below is a point-by-point response to the reviewers' comments and suggestions.

Comments:

1. The initial review raised the following point:

Also missing from the first section is any description of how the data was split for training, validation, and testing. In perusing the Methods section, I discovered a potentially serious problem with the proposed approach, namely, that the data is based on a random split of 200bp windows. This is problematic because, as stated on line 90, the features include 1 kb of context around each window. The result is that if two adjacent windows get sent to the train and test sets, then the training set will contain features from the test set. This is exactly the mistake that Katie Pollard's group made in training a promoter-enhancer predictor, which led to this critique: <https://nam04.safelinks.protection.outlook.com/?url=https%3A%2F%2Fwww.nature.com%2Farticles%2Fs41588-019-0434-7&data=05%7C01%7Cchengqi%40usf.edu%7Ca099b2eb047f4f60f9b608db678a3fe5%7C741bf7dee2e546df8d6782607df9deaa%7C0%7C0%7C638217615713589558%7CUnknown%7CTWFpbGZsb3d8eyJWljiMC4wLjAwMDAiLCJQIjoiV2luMzliLCJBTiI6IjEhaWwiLCJXVCi6Mn0%3D%7C3000%7C%7C&sdata=ex2Kg8RZe%2FnzKqzW5J4SMzR4Aoj%2BI0MyxcGurJAeEMI%3D&reserved=0>

I found the authors' response unsatisfactory. There are two ways to approach this point:

1. Perform testing on left-out chromosomes, which is the most robust way of performing this kind of analysis.
2. Perform an analysis of the potential overlap between peaks. The authors state that this is likely to be low due to the sparsity of ChIP-seq peaks in *P. falciparum*, but fail to provide quantitative data to support this claim. Instead they state that "While split by chromosomes can effectively separate unwanted information sharing, it misses the opportunity to study the effects in those left-out chromosomes. Our use of random split is a trade-off for less bias over less variance." There are two false claims here. First, if they really want to provide test results on the entire genome, they can perform a cross-validation analysis. Even with their current analysis they are missing lots of peak regions from testing. Second, the term "bias" here is mis-used and demonstrates a lack of understanding of the bias/variance tradeoff in machine learning. Infact, their results on left out chromosomes as suggested by the reviewer, suggest that this is indeed an issue.

	Test set			Validataion set		
	#overlap region(50%-80%)	#regions	Fraction of overlaped regions	#overlap region(50%-80%)	#regions	Fraction of overlaped regions
P. falciparum						
H3K9ac_Trophozoites	1170	1562	75%	1199	1562	76%
H3K9ac_Schizonts	66	146	45%	68	146	47%
PfBDP1_Trophozoites	27	35	77%	29	35	83%
PfBDP1_Schizonts	133	163	82%	128	163	78%
AP2I_Schizonts	30	48	62%	27	48	56%
AP2G_Schizonts	26	57	46%	36	57	63%
AP2G_Sex_rings	21	40	52%	23	40	57%
AP2G_Gametocytes	91	136	69%	82	136	60%
AP2G5_Trophozoites	184	333	55%	171	333	51%
ATAC_Seq_05_10h	66	146	45%	68	146	46%
ATAC_Seq_15_20h	134	267	50%	133	267	50%
ATAC_Seq_25_30h	563	883	64%	564	883	64%
ATAC_Seq_35_40h	539	845	64%	510	845	60%
P. berghei						
AP2G2_Gametocytes	502	927	54%	530	927	57%
AP2O_Ookinetes	220	342	64%	208	342	61%

Table 1. We assessed the proportion of 1KB regions that exhibited overlap with either the testing or validation set. We considered only those regions with a minimum of 50% overlap. Given that the model development involved selecting 200bp genetic regions and extending them to 1kb length, the maximum observed

We express our gratitude to the reviewer for highlighting this question and offering potential solutions. In response to the reviewer's suggestion, we have thoroughly examined the overlapping 1kb region between the training and test sets (point 2 from the reviewer 3). Furthermore, we have implemented the left-out chromosome strategy while training our model (point 1 from the reviewer 3). In our initial analysis, we measured the proportion of 1kb genomic regions

that exhibited overlap between the training and test sets, as well as with the validation set. As anticipated by the reviewer, a substantial fraction of 1kb genomic regions showed overlap between the training set and either the test or validation set, as indicated in Table 1. Consequently, we adopted a chromosome splitting strategy to train our model. This involved grouping 1kb regions on the same chromosome together, assigning them to either the training, validation, or test set. We select chromosome combinations in the training, validation, and test set so that the ratio is close to 7/1.5/1.5 for the chromatin profiles with low positive samples (positive bin number < 1,100). For the remaining chromatin profiles, we designated bins on chromosomes 10 and 11 as the validation set and bins on chromosomes 9 and 12 as the test set. All bins in the remaining chromosomes were used to form the training set, as outlined in Supplement Table 3 in the manuscript. The description for this part can be found on Page 16, lines 31 to 40. After comparing with the random splitting strategy, we observed an average decrease of 0.03 in auROC and 0.14 in average auPRC. In the discussion section, we discussed the distinctions between the random splitting and chromosome splitting strategies on Page 15, line 33 to 48.

2. In the introduction the authors note that

"Sequence-based prediction models provide a new perspective on the effects of genomic variants on gene regulation. However, the majority of available models were developed in model organisms, which are not suitable for malaria parasites due to several unusual features of their genome architecture"

As mentioned in the initial review, this statement is inaccurate. The authors use existing model architectures very similar to those used in model organisms, contradicting what they have said. I suggest rewording as follows:

"Sequence-based prediction models provide a new perspective on the effects of genomic variants on gene regulation. However, the majority of available models were developed in model organisms, and their applicability to malaria parasites whose genomes have several unusual features, requires careful evaluation."

In this context the authors note that "the applicability of existing predictive models include the

apparent absence of some subunits of the preinitiation complex". I don't see how this is case - please explain how that changes anything about the modeling approach.

We appreciate the reviewer's suggestion to rephrase the sentence. The sentence in question has been replaced with the specific suggested sentence from the reviewer (Page 3, line 11 to 12). Furthermore, we acknowledge that mentioning the "preinitiation complex" in that context may be inappropriate. Since our focus is on abnormal genome properties in malaria parasites related to DL training for epigenetic profile prediction, the differences in the preinitiation complex compared to model organisms may not be directly relevant to this topic. We already remove this in line of the manuscript (Page 3, line 17-18).

3. "This framework is complementary to experimental approaches at higher throughput with lower cost". While it's true that it is complementary to experimental approaches, it's not meant to replace them - you need the experimental data to be able to perform this type of analysis. Rather, your framework enables to make the most out of the existing experimental data and derive biological insight from it (e.g. effect of non-coding variants).

We agree with the reviewer's comment.

In our revision, we have modified it to 'The framework complements experimental approaches by providing higher throughput and lower cost. However, it is not intended to replace them, as its performance relies on experiment results for training. Instead, this framework enables us to maximize the potential of existing experimental data and extract valuable biological insights from it' (Page 3, line 38-42).

4. "The unusual genome property (e.g., ~80% AT content on average, rising to ~92% in intergenic regions) of malaria parasites requires the development of a novel computational framework to understand the regulatory sequence code from parasite noncoding sequences."

The proposed framework is not novel; I suggest rephrasing as follows:

The unusually high AT content of the P falciparum genome (80% AT on average, rising to ~92% in intergenic

regions) requires careful evaluation of existing deep learning approaches for understanding the regulatory sequence code from parasite noncoding sequences.

We acknowledge the reviewer's comment and have revised the relevant section on Page 3, lines 48 to 50, of the manuscript accordingly.

5. "Here, s is an input sequence represented as a $4 \times \text{length}$ matrix, where each row represents a nucleotide, and each column indicates a position in the length sequence. '1' represents a specific nucleotide present at a specific position of the sequence, and '0' does not (Fig. 1A)." Suggest replacing with:

"Here, s is an input sequence represented as a $4 \times \text{length}$ matrix, where each column represents a position in the sequence with a "1" in the row corresponding to the nucleotide at that position, and has zeros otherwise."

We agree the suggestions provided by the reviewers, and we have already made the change on Page 4, line 23-25.

6. "The convolution layer (Conv) is a motif scanner analogizing the position weight matrix." I think you meant that

The convolution layer (Conv) functions as a motif scanner analogous to a position weight matrix.

We agree the suggestions provided by the reviewers, and we have already incorporated the necessary changes into our revision, specifically on Page 4, line 29-30.

7. "The LSTM is crucial since TF binding sites may have a group of sequence patterns separated from each other by a long distance" You provide justification for this statement later, but should be referred to here as well.

We added the sentence "we showcased the significance of LSTM by substituting it with a convolutional layer, and the subsequent discussion regarding this comparison can be found in the following paragraph" (Page 4, line 35-38).

8. The new section "MalariaSED prediction in malaria parasites outperforms state-of-the-art frameworks"

developed in model organisms" is poorly conceived. The good part is that it demonstrates the advantage of using the LSTM layer. However, I don't think that the ENFORMER architecture comparison makes a valid statement. The ENFORMER framework is designed for much larger datasets such as those that one finds in large genomes like the human genome. Therefore, I do not find it surprising that it did not perform very well, and don't see the point of reporting on such a failed experiment. Here you are opening yourselves to the criticism that perhaps it didn't work because you did not optimally select its hyperparameters. The contribution of your work is on demonstrating the ability of DL to work well in this interesting genome, using commonly used DL techniques. I suggest simply reporting the fact that LSTM provided improved performance over a purely convolutional approach in your experiments as part of the previous section where the statement about the contribution of the LSTM to performance was made. I would also like to note that the purely convolutional approach you compare to was state-of-the-art in 2015 which is ages ago when it comes to developments in DL. Therefore the statement that MalariaSED outperforms state of the art frameworks is not really valid.

We appreciate the valuable comments provided by the reviewers and have taken them into careful consideration. The inclusion of Enformer in our study was in response to the request from review 2, and we did not optimize its hyperparameters. To address this, we have removed the Enformer results from the method section and confined the discussion part about Enformer in Page 15, line 24 to line 31. In response to the reviewer's suggestion, we have included the demonstration of LSTM performance improvement over the solely convolutional layer in this paragraph. Additionally, we have updated the section title to 'The LSTM layer enhances the performance of MalariaSED in comparison to using solely convolutional layers' to reflect this change (Page 5, line 16).

9. "We also tested the MalariaSED performance on chromosome splitting strategy with average auROC = 0.93 and auPRC = 0.59 (Supplement Table 3)."

This statement does not correctly represent the fact that this is the preferred strategy. Please rephrase e.g. as:

We also tested the MalariaSED performance on a chromosome splitting strategy with chromosomes XX as the test set, achieving an average auROC = 0.93 and auPRC = 0.59 (Supplement Table 3). We note that this is the preferred evaluation strategy as it avoids biases that result from potentially overlapping examples (see e.g. <https://nam04.safelinks.protection.outlook.com/?url=https%3A%2F%2Fwww.nature.com%2Farticles%2Fs41576-021-00434-9&data=05%7C01%7Cchengqi%40usf.edu%7Ca099b2eb047f4f60f9b608db678a3fe5%7C741bf7dee2e546df8d6782607df9deaa%7C0%7C0%7C638217615713589558%7CUnknown%7CTWFpbGZsb3d8eyJWljoic4wLjAwMDAiLCJQIjoiV2luMzliLCJBTiI6Iik1haWwiLCJXVCi6Mn0%3D%7C3000%7C%7C%7C&sdata=ceJ8NiqZrqf8FAywFlbAqxEbFbYxTwNlfpmX8S%2Fv9U%3D&reserved=0>).

This educates the readers on the proper way of running their experiments, and demonstrates that it does have an effect.

We are grateful for the valuable comments provided by the reviewer. We have revised this section accordingly, and you can find the updated version on Page 4, line 45 to 49, in the manuscript.

10. In the first section of the Results section I would like to see a bit more information about what you are predicting:

There is a single sentence stating that you are using "large-scale epigenetic experimental data, including open chromatin accessibility (ATAC-Seq)²⁹, TF binding, and histone-mark profile (ChIP-Seq)" I would have liked to see more information about how many overall profiles are being predicted (the number appears later) and which TFs and histone-marks are being profiled. As is, it is lacking concreteness.

In response to the reviewer's feedback, we have included an additional sentence.

The data includes four open chromatin accessibility (ATAC-Seq)²⁸ profiles across *P. falciparum* intraerythrocytic development cycle (IDC), six transcription factors (TFs) with their nine binding profiles during *P. falciparum* and *P. Berghei* IDC and sexual stages, as well as two H3K9ac histone-mark profiles in *P. falciparum* mid and late IDC stage' on Page 4, line 17 in the revision.

11. I would have liked to see that your model is able to learn binding motifs that are similar to the known motifs of ApiAP2s transcription factors.

In light of the reviewer's feedback, we employed MalariaSED to compute the transcriptional factor binding effects of all potential sequences sharing the same length as the motifs under investigation. We included the average binding effects for each of these sequences in supplement table 5 in the revised version. Notably, the previously identified motifs (AP2G, AP2I, AP2G2, AP2O, and PfBDP1, Figure 4 in revision) consistently emerged as the top-ranked ones, exhibiting the most potent binding effects when subjected to single nucleotide substitution.

12. "Single-nucleotide substitutions at the TF motif weaken TF binding" This is exactly what we would expect, since the convolutional layer is a collection of motif detectors! This is a standard observation in studies that use DL for genomics data.

Indeed, we acknowledge and appreciate the reviewer's comments. As the malaria parasite research community is predominantly composed of wet lab researchers, it is crucial to validate that the predictions derived from DL models align with existing knowledge, especially regarding known motif patterns already established in the parasite research community.

To address this necessity, we utilized MalariaSED to assess the TF binding effects resulting from single nucleotide substitution. This approach allowed us to gain insights into the contributions of all sequence patterns sharing the same length as the reported motifs to the investigated TF binding. By employing MalariaSED, we aimed to demonstrate the consistency of DL-based predictions with the prior knowledge derived from the well-established motif patterns in malaria parasite research.

13. The previous review noted that the Discussion re-iterates many points introduced in the Results section and do not add to the manuscript. The discussion needs to contain NEW insights regarding the results rather than a rehashing of what you already wrote. Furthermore, my gripes with the discussion of your Enformer results hold even more for what you wrote in the Discussion.

We thank reviewer's comment. We have removed the majority of discussion part regarding Enformer and emphasis the necessary of further work to exploring the application of it in malaria parasites. We also removed the majority discussion regarding comparisons with convolutional network, and mentioned the further effort should optimize the kernel size for comparisons with MalariaSED. The majority of our discussion focuses on explaining the discrepancies between

MalariaSED predictions and experimental results, as well as providing insights into the chromatin effects of genetic variants. Many of these explanations were not included in the results section. For instance, we explored the potential reasons behind noncoding variants showing high levels of geographic differentiation and their likely impact on surrounding epigenetic profiles. These impacts could arise from natural selection or deleterious changes that have spread to high frequency due to hitchhiking with selected coding variants. While some aspects of our discussion reiterate certain observations, we consider this repetition important as it helps refresh the audience's understanding of our findings. Furthermore, these explanations help to shed light on any discrepancies between our predictions and experimental results, adding depth to the interpretation of our research outcomes.

14. In the purely convolutional network you used hyperparameters similar to those used to analyze the human genome. This is despite the fact that the malaria genome is much smaller, and that you modeled a dataset consisting of a lot fewer experimental datasets. So this model was not provided with the opportunity to do well, so obviously did not perform as well as the LSTM based version.

We appreciate the feedback provided by the reviewer. We indeed employed the same kernel size as the published DL model in the human genome. However, it is essential to clarify that the first and second convolutional layers in our model share the same kernel number as MalariaSED, with the key distinction being the third layer, which utilizes a convolutional layer instead of an LSTM layer. Regarding hyperparameter optimization, we thoroughly explored and optimized all relevant parameters. The details of our hyperparameter search space can be found in supplement table 4.

In the revised version, we have toned down the comparisons with the previously established DL model. In addition, we have emphasized the need for future research to explore and investigate various kernel sizes within the convolutional network.

15. In your svm experiments you used k-mers that are very long. In fact, the original paper you cite mentions that "predictive power begins to decrease when k is greater than six. You used values of seven and above. A better SVM method to compare to is the GKM-SVM method that implements a more flexible kernel method.

	AUC_ROC	AUC_PR
ATAC_Seq_05_10h	0.69	0.17
ATAC_Seq_15_20h	0.67	0.12
ATAC_Seq_25_30h	0.70	0.21
ATAC_Seq_35_40h	0.69	0.26

We are grateful for the reviewer's feedback, and in response, we employed GKM-SVM with 6-mer sequence input. We maintained the same procedure for negative set extraction as in the first round revision, ensuring a matched distribution of sequence length, repeat

element fraction (each k-mer sequence pattern frequency), and GC content with the corresponding positive set. The training and test sets were randomly generated in a ratio of 8.5/1.5. For model training, we used the default parameters of GKM-SVM, which involved 10-fold cross-validation, and we present their performance on the test set. While we observed some improvement with the SVM (Lee et al., Genome Res, 2011, PMID: 21875935) used in our previous revision, it still exhibits limited performance compared to DL frameworks in discriminating ATAC-Seq peaks.

16. The authors mention that their method was trained using histone-mark profile data, but no other information is provided in this matter - which histone-mark, where did the data come from, and what was its contribution to network performance.

We appreciate the reviewer's feedback. We apologize for the oversight in not including the histone marker used in MalariaSED training in the method section, although we did mention it at the beginning of the results. We have now rectified this issue in the revised version. The relevant information about the histone marker, which is H3K9ac in trophozoites and schizonts, has been added to the method section (Page 16, line 22).

17. Very few TFs are known in the malaria parasite as noted by the authors. This could be the result of poor annotations rather than an organism with very few TFs. Given that the network was trained on chromatin accessibility data, it might have picked up signals from TFs that are not represented in the TF- ChIP-seq datasets. Strong motifs that do not match to known ApiAP2 motifs are candidates for novel TFs. This could be a good avenue for further research. MalariaSED primarily underwent training using the *P. falciparum* genome. Being the most lethal malaria parasite, the genome annotation of *P. falciparum* stands out as significantly more comprehensive compared to other apicomplexa parasites. In *P. falciparum*, only 73 TFs were identified, despite the genome containing more than 5000 genes. This number is significantly lower compared to other organisms, such as yeast with approximately 170 TFs for 5400 genes, and human cells with over 1500 TFs for 20,000 genes. The *P. falciparum* transcription factors can be categorized into eight helix-turn-helix proteins, 37 C2H2-type zinc fingers, and one β -scaffold factor. However, it is worth noting that these factors exhibit only limited conservation across different *Plasmodium* species (PMC2821373, PMC6859821). In contrast, the ApiAP2 family, presumed to be the primary regulators of transcription in the parasite's life cycle (PMC1178005, PMC509263), displays greater significance and relevance across various *Plasmodium* species. Given the limited experimental results presented in our manuscript, we are cautious about challenging the well-established consensus in the malaria research community that 'there are very few TFs in the *P. falciparum* genome'.

However, the experimental validated TF binding profiles in *P. falciparum* is extremely limited, posing a significant hindrance to the discovery of DNA binding motifs. We acknowledge the reviewer's comment regarding the novel motif identified from open chromatin accessibility data, which could not represent any known TF binding motifs. This has been confirmed in the original ATAC-Sea results, which also revealed the existence of multiple novel motifs and several known motifs in other organisms (resulting in a total of ~35 motifs first discovered in the *P. falciparum* genome, PMC5899830). Given that MalariaSED was trained using this experimental data, and a substantial number of motifs have already been identified and their functions discussed in the original ATAC-Seq publication, we are cautious about mentioning any novel motifs identified in our established DL model without experimental validation

Typos and grammar

"experimental data, including open chromatin accessibility (ATAC-Seq), TF binding, and histone-mark profile  profiles

We have added 'S' after the word 'profile' as instructed.

"We use early stopping to seize the parameter optimization process if there is no improvement in validation auPRC for ten consecutive epochs." replace seize with "halt"

We have already replaced 'seize' with 'halt'.

The reference "Practical Bayesian Optimization of Machine Learning Algorithms" was published in the Neurips conference. Also, the method itself is called

We have added the correct reference as required. Thanks for bringing this to our attention

"with the objective to maximum auPRC"  objective to maximize

We have eliminated the phrase 'with the' from the text.

The ROC and PR curves were generated based on the MalariaSED prediction from the sigmoid activation function...  from the network output...

We have updated the word to 'from the network output' as per the change requested